# Porphyrin overdrive rewires cancer cell metabolism

Swamy R Adapa[1,7,]*, Gregory A Hunter[2,]*, Narmin E Amin[3], Christopher Marinescu[1], Andrew Borsky[2], Elizabeth M Sagatys[4], Said M Sebti[5], Gary W Reuther[3], Gloria C Ferreira[2,6,7], Rays HY Jiang[1,7]

All cancer cells reprogram metabolism to support aberrant growth. Here, we report that cancer cells employ and depend on imbalanced and dynamic heme metabolic pathways, to accumulate heme intermediates, that is, porphyrins. We coined this essential metabolic rewiring "porphyrin overdrive" and determined that it is cancer-essential and cancer-specific. Among the major drivers are genes encoding mid-step enzymes governing the production of heme intermediates. CRISPR/Cas9 editing to engineer leukemia cell lines with impaired heme biosynthetic steps confirmed our whole-genome data analyses that porphyrin overdrive is linked to oncogenic states and cellular differentiation. Although porphyrin overdrive is absent in differentiated cells or somatic stem cells, it is present in patient-derived tumor progenitor cells, demonstrated by single-cell RNAseq, and in early embryogenesis. In conclusion, we identified a dependence of cancer cells on non-homeostatic heme metabolism, and we targeted this cancer metabolic vulnerability with a novel "bait-and-kill" strategy to eradicate malignant cells.

## Introduction

Heme biosynthesis is one of the most efficient metabolic pathways in humans. A total of 270 million molecules of heme are produced for every RBC (D'Alessandro et al, 2017), at a rate of over 2 million RBCs per second (Dean, 2005). Heme, a porphyrin ring encaging most of the iron in humans, functions as an essential prosthetic group of numerous proteins with roles ranging from signal sensing, DNA binding, microRNA splicing and processing to enzymatic catalysis (Ponka et al, 2014). To ensure the enormous output of heme biosynthesis, the supply of substrates, intermediates, and end products in the pathway is tightly regulated and precisely balanced (Hunter & Ferreira, 2011). Heme is a double-edged sword for cell growth; it is essential in the "right amount" (Cao & Dixon, 2016) and if

not can be toxic (Malik & Djaldetti, 1980) via unique forms of cell death.

Cancer cells reprogram metabolic pathways to fuel anabolic growth, rapid propagation, and efficient nutrient acquisition (Chaffer & Weinberg, 2011; Martinez-Outschoorn et al, 2017). Although some studies have implicated heme in carcinogenesis through cytotoxic heme-derived compounds (Malik & Djaldetti, 1980), lipid peroxidation (Martin et al, 2018), oxidative damage (Sohoni et al, 2019), intestinal flora toxicity (Ijssennagger et al, 2015), and energy production (Fiorito et al, 2021), little is known about how heme biosynthesis deregulation and heme trafficking alterations contribute to tumor dependence on heme and its biosynthetic pathway for survival. Here, we describe "porphyrin overdrive," a novel cancer cell metabolic reprogramming characterized by imbalanced and dynamic heme metabolic pathways essential for cancer cell growth.

"The Warburg effect" (Vander Heiden et al, 2009) and "glutamine addiction" (Shelton et al, 2010) are two altered forms of metabolism found in cancer cells. Therapeutic interventions based on these altered cancer metabolic pathways are challenging because glycolysis and glutamine metabolism are required in every cell, and thus, such therapeutic approaches lead to non-specific toxicity. In contrast, we propose that porphyrin overdrive is an ideal cancer metabolic pathway for therapeutic targeting as we demonstrate that it is (1) cancer cell–essential (i.e., cancer cells require it for survival) and (2) cancer-specific (i.e., it is absent in normal cells).

## Results

### Cancer is characterized by aberrant heme metabolism

Compared with normal cells, cancer cells have enhanced metabolic dependencies (Levine & Puzio-Kuter, 2010; Luengo et al, 2017; Martinez-Outschoorn et al, 2017). Our initial data analyses of genome-scale CRISPR/Cas9 gene loss of function from the publicly available project DepMap screens of cell lines derived from a set of

---

[1]USF Genomics Program, Center for Global Health and Infectious Diseases, College of Public Health, University of South Florida, Tampa, FL, USA [2]Department of Molecular Medicine, Morsani College of Medicine, University of South Florida, Tampa, FL, USA [3]Department of Molecular Oncology, H. Lee Moffitt Cancer Center and Research Institute, Tampa, FL, USA [4]Department of Pathology, H. Lee Moffitt Cancer Center and Research Institute, Tampa, FL, USA [5]Department of Pharmacology & Toxicology, Massey Cancer Center, Virginia Commonwealth University, Richmond, VA, USA [6]Department of Chemistry, College of Arts and Sciences, University of South Florida, Tampa, FL, USA [7]Global and Planetary Health, College of Public Health, University of South Florida, Tampa, FL, USA

Correspondence: jiang2@usf.edu
*Swamy R Adapa and Gregory A Hunter contributed equally to this work

---

  

cancer cell lines derived from diverse tissue origins (Meyers et al, 2017) indicated that cancer cells depend on heme synthesis (Fig S1) (Table S1). Significantly, these cancer cell lines developed dependencies on heme metabolism–related proteins, such as uroporphyrinogen III decarboxylase (UROD), the enzyme that catalyzes the fifth step in the heme biosynthetic pathway. These cancers also depend on a set of hemoproteins, such as cytochrome c-1 (CYC1), succinate dehydrogenase complex subunit C (SDHC) for cellular respiration, and DGCR8 for microRNA biogenesis (Nguyen et al, 2015), each of which uses heme as a co-factor. Unexpectedly, the ferrochelatase (*FECH*) gene encoding FECH that catalyzes the final critical step of heme biosynthesis is dispensable in several cancers, suggesting that cancer cell lines are capable of bypassing endogenous biosynthesis of heme in vitro, yet are still dependent on genes encoding enzymes that mediate intermediate steps of heme biosynthesis. This initial analysis of essentiality data indicated a greater dependency of these cancer cell lines on the mid-steps of biosynthesis genes, as opposed to the last step of the pathways. This observation suggests that the cancer pathways accumulate intermediates more than the end product, which is indicative of an "inefficient" pathway because of imbalanced synthesis enzyme steps. This stands in contrast to normal heme biosynthesis, which is well balanced, with no intermediate accumulation, and efficient conversion of substrates into products (Fig 1A).

Next, we investigated the genes that encode proteins acting as gatekeepers for heme biosynthesis. The first committed and key regulatory step in mammalian heme biosynthesis is catalyzed by 5-aminolevulinic acid (ALA) synthase (ALAS; EC 2.3.1.37) (Ponka et al, 2014). Two chromosomally distinct genes, *ALAS1* and *ALAS2*, encode the housekeeping and erythroid-specific ALAS isoforms, respectively (Ponka et al, 2014). Although *ALAS1* is expressed in every cell, the expression of *ALAS2* is restricted to developing erythrocytes (Ponka et al, 2014). We examined the data generated from genome-wide CRISPR-based screens of diverse cancer cell lines, which were designed to provide a complete gene essentiality dataset for these cell lines (Meyers et al, 2017; Tsherniak et al, 2017). In contrast to *ALAS2*, the essentiality of *ALAS1* is a feature of diverse cancer cells independent of cancer type (Fig S2A). Consistent with these forward genetic results, the gene expression patterns in ~10,000 patient tumors from the Genotype-Tissue Expression (GTEx) (Lonsdale et al, 2013; The GTEx Consortium et al, 2015) and The Cancer Genome Atlas (TCGA) (Cancer Genome Atlas Research Network et al, 2013) datasets show that *ALAS2* expression is absent in most tumors except myeloid leukemias (Fig S2B). These results show that despite ALAS2 being responsible for most heme production (~85%) in humans, many cancer cell lines, including erythroleukemia of RBC lineages (e.g., HEL), require ALAS1 (Fig S2A–C).

## Porphyrin overdrive operates in diverse cancers and early embryos

To better understand the molecular and genetic mechanisms underlying the cancer dependency on heme metabolism, we determined gene essentiality from genome-scale CRISPR/Cas9 loss-of-function screens in over 300 human cancer cell lines covering different cell lineages and estimated gene dependency (Meyers et al, 2017; Tsherniak et al, 2017). Then, we focused on the gene

dependencies associated with the eight enzymatic steps of the heme biosynthetic pathway by computing the respective pan-cancer essentiality scores, similar to the lethality scores used in the development of a cancer dependency map (DepMap) (Dempster et al, 2019; Pacini et al, 2021). Gene essentiality score values lower than zero indicate decreased cell growth/viability with loss of gene X, and hence, the lower the X gene essentiality score, the more dependent are cells on the X gene. Consistent with recent cross-platform studies (Dempster et al, 2019; Pacini et al, 2021), over 2,000 genes were defined as pan-cancer–essential, based on the distribution of their essentiality scores in over 300 cancer lines derived from 27 of the most prevalent cancer types. The gene essentiality scores were confirmed by using the same method on the total DepMap collection (DepMap release 21Q3) of over 1,000 cell lines (Pearson's R > 0.9, *P* < 0.001). Strikingly, instead of finding the expected complete and balanced heme biosynthetic pathway characteristic of normal cells, we found that the survival of cancer cells depends on the various steps of the heme biosynthetic pathway in a "non-balanced" manner. In 100 different types of cancers, the tumor cells particularly rely on the intermediate enzymatic steps of heme biosynthesis, as assessed by loss of cellular viability upon CRISPR/Cas9-mediated gene ablation in whole-genome loss-of-function screens (Fig S3) (Tables S2 and S3). Specifically, as shown in Figs 1B and S3, the pan-cancer gene essentiality is the highest for the genes encoding the enzymes responsible for the fifth and sixth steps of the pathway, UROD and coproporphyrinogen III oxidase (CPOX), respectively. This result was notably surprising, as the genes for the ALAS isoforms, which catalyze the initial and rate-limiting step in the pathway, and the gene for FECH, the final enzyme in the pathway that catalyzes heme production, had lower essentiality values and were dispensable in several cancers (Figs 1B and S3). Though the heme biosynthesis genetic dependences of 27 major forms of human cancer vary, the estimated patterns of essentiality across the cancer lines clearly revealed a range of heme and heme precursor requirements. These results uncovered the "unbalanced" nature of heme biosynthesis in cancer, with many cancers surviving without functional first and terminal enzymatic steps but depending on the intermediate steps (Fig S3). The gene essentiality analysis also highlighted the importance of heme trafficking (e.g., heme importer FLVCR2) and key hemoproteins (e.g., CYC1) in cancer cell survival (Figs 1B and S3).

To examine cancer heme metabolism in vivo, we analyzed the in vivo mouse CRISPR/Cas9 loss-of-function and metabolic essentiality data from pancreatic and lung cancer models (Zhu et al, 2021). The genetic screen (Zhu et al, 2021) indicates that most metabolic gene essentialities were similar between the in vitro and in vivo settings, except for the heightened requirement of the intermediate steps of heme biosynthesis in vivo. As reported in the previous study, of the ~2,900 studied metabolic genes (Zhu et al, 2021), the genetic dependencies of both murine pancreatic and lung cancers were only increased for the enzymes responsible for the intermediate steps of the heme biosynthetic pathway (Fig S4A and B) (Table S4). Strikingly, our careful examination of each step of the heme biosynthetic pathway shows that the essentiality extent of either *ALAS* or *FECH*, which both code for the first and terminal enzymes of the heme biosynthetic pathway, respectively, did not differ significantly between the in vivo murine cancer models.

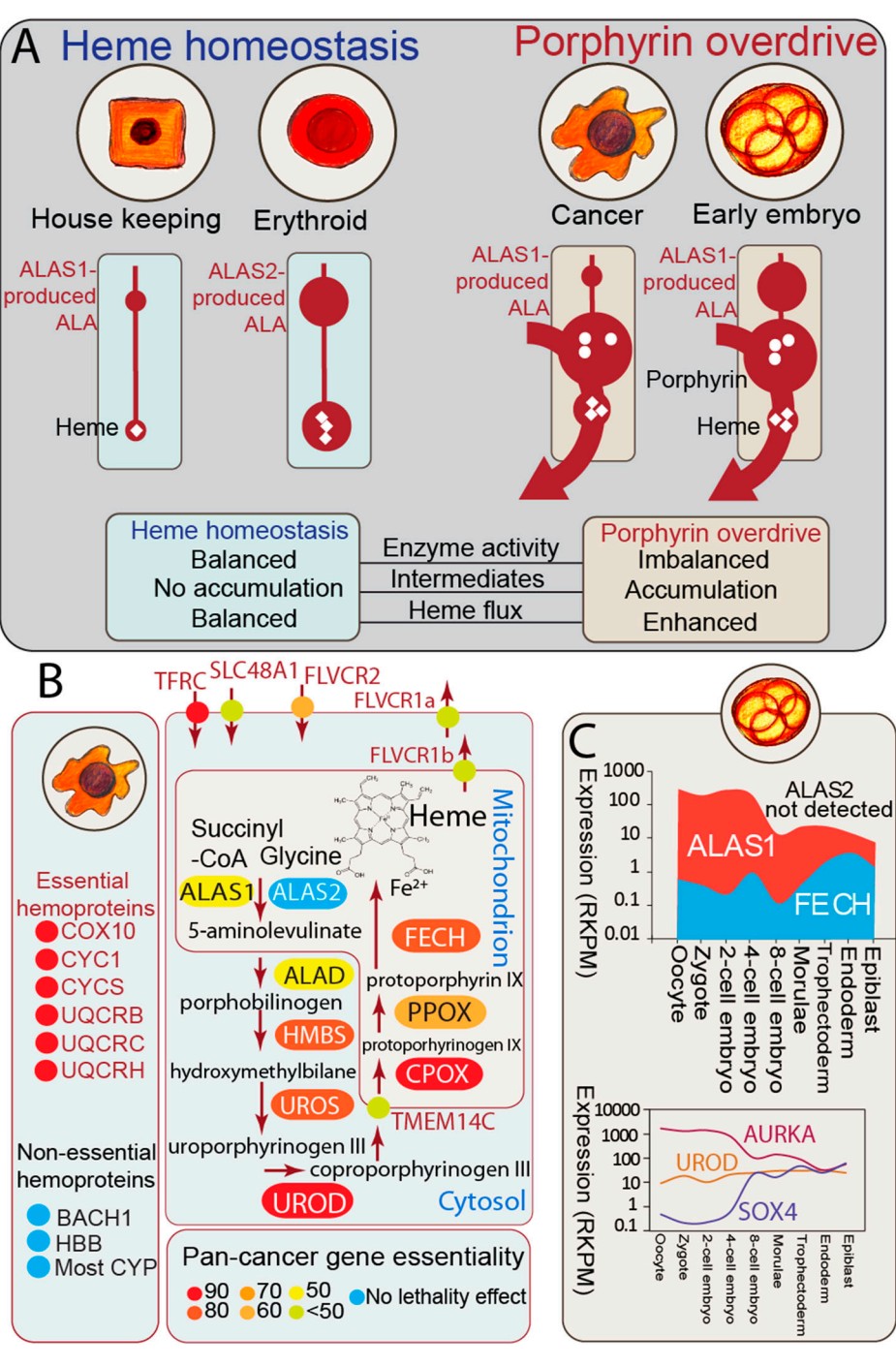

**Figure 1. Defining porphyrin overdrive with CRISPR data and early human embryonic stem cell single-cell RNAseq.**
**(A)** Schematic illustration of heme homeostasis and porphyrin overdrive. Heme homeostasis refers to the heme biosynthetic pathway in normal cells, with the housekeeping enzyme ALAS1 catalyzing the first and rate-limiting step in all cells, except precursor erythroid cells, where the erythroid-specific ALAS isozyme ALAS2 catalyzes the rate-determining step. Porphyrin overdrive refers to the imbalanced (i.e., with aberrantly increased and suppressed enzyme activities) pathway in cancer cells and early human embryonic stem cells, with the ALAS1 isozyme controlling ALA production regardless of the cell type. White circles and white diamonds represent porphyrin intermediates and heme, respectively; erythroid refers to an erythroid precursor cell. Imbalanced enzyme activities in porphyrin overdrive are indicated by different sizes of product pools (dark red circle sizes). ALA, 5-aminolevulinate; ALAS, ALA synthase. **(B)** Defining porphyrin overdrive with CRISPR KO–derived essentiality data in over 300 cancer cell lines. Analysis of CRISPR essentiality data enables mapping of porphyrin overdrive in 27 major types of cancer. Color mapping indicates average cancer cell growth dependence revealed by CRISPR KO of a given gene. UROD has the highest pan-cancer essentiality. **(C)** Early human embryonic stem cells show features of porphyrin overdrive. The expression levels for *ALAS1* and *FECH* differ over 100-fold during the first rounds of embryonic stem cell divisions, but they normalize with the early embryonic development progression. AURKA and SOX4 show expected patterns of expression dynamics during embryogenesis (abbreviations: ALA, 5-aminolevulinate; ALAS1, housekeeping ALA synthase; ALAS2, erythroid-specific ALA synthase; ALAD, ALA dehydratase [aka porphobilinogen synthase]; BACH, transcription regulator BACH [BTB and CNC homology]; CPOX, coproporphyrinogen oxidase; COX10, cytochrome C oxidase assembly homolog 10; CYC1, cytochrome C1; CYP, cytochrome P450 family; CYCS, cytochrome c, somatic; FECH, ferrochelatase; FLVCR, feline leukemia virus subgroup C receptor family; FLVCR1a, FLVCR member 1a; FLVCR1b, FLVCR member 1b; FLVCR2, FLVCR member 2; HBB, hemoglobin β-subunit; HMBS, hydroxymethylbilane synthase; PPOX, protoporphyrinogen oxidase; RKPM, reads per kilobase million; TFRC, transferrin receptor; TMEM14C, transmembrane protein 14C; UROD, uroporphyrinogen decarboxylase; UROS, uroporphyrinogen III synthase; UQCRB, ubiquinol–cytochrome c reductase binding protein; UQCRC, ubiquinol–cytochrome c reductase core protein 1; UQCRH, ubiquinol–cytochrome c reductase hinge protein).

Although the Zhu study interprets these results as enhanced heme production in vivo, our close analysis of the linear heme biosynthetic pathway reveals that, in fact, these CRISPR KO results show the significance of heme intermediates and the imbalanced pathway's role in intermediate accumulation within an in vivo setting (Fig S4).

To assess heme biosynthesis in patient tumor samples, we analyzed data available from the GTEx project (Lonsdale et al, 2013; The GTEx Consortium et al, 2015) and TCGA program (Cancer Genome Atlas Research Network et al, 2013). We used methods designed for the pairwise comparative gene expression of GTEx and TCGA datasets (Tang et al, 2017). In over 80% of all tumors, the genes for hydroxymethylbilane synthase (HMBS), the third enzyme of the heme biosynthetic pathway, and the heme exporter FLVCR1 were among the most commonly up-regulated across diverse tumor types (Figs S5A and B and S6) (Table S5). By the same criteria, the

onco-signaling genes *AURKA*, *KRAS*, and *MYC* were up-regulated in over 90%, 60%, and 50% of all tumor types, respectively. In contrast, the expression of the genes encoding the second and terminal enzymes of the heme biosynthetic pathway, aminolevulinate dehydratase (a.k.a. porphobilinogen synthase) and FECH, respectively, is down-regulated, suggesting the presence of a buildup of intermediate heme precursors in tumors. *HPX*, the gene for the heme-binding and scavenger hemopexin (Tolosano et al, 2010), was also down-regulated in most of the tumors (Fig S5C). This finding, which corroborates the previously reported role of hemopexin as a key player for the checkpoint in cancer growth and metastases (Canesin et al, 2020), leads, compared with the gene expression patterns in ~10,000 tumors versus those in normal tissues, us to suggest that the heme content is high in the tumor microenvironment. *ALAS2* belongs to the 10% of genes that were not expressed in most of the large tumor collections, except myeloid leukemias (<2% of total tumors). Overall, our comparative analysis of the differential gene expression in tumor tissues, spanning 31 cancer types, indicated that only a subset (e.g., third step) of the nine genes encoding the enzymes of the heme biosynthetic pathway were up-regulated in cancer cells, whereas the second-step or the last-step genes in fact are often down-regulated in tumors.

Based on the gene essentiality results from both in vitro (Figs 1B and S3) and in vivo (Fig S4) settings, the pan-cancer gene expression patterns (Figs S5 and S6), the detected porphyrin accumulation in a wide range of tumors (Fiorito et al, 2020), and the reported elevated heme flux in diverse cancer cells (Hanna et al, 2016), we defined the salient porphyrin overdrive–associated features. They are as follows: (1) heme biosynthetic enzymes with aberrantly increased (HMBS, UROS, and UROD) and suppressed (aminolevulinate dehydratase and FECH) activities, (2) accumulation of heme precursors (porphyrins), and (3) enhanced heme flux (Fig 1A and B). This cellular status contrasts with the tightly regulated steady state or homeostasis, defined by a flawless enzyme-catalyzed channeling of substrates to products along the heme biosynthetic pathway and circumventing the toxicity and instability of the pathway intermediates, while ensuring that the cell heme requirements are met (Ferreira, 2013). With porphyrin overdrive, the production of heme precursor molecules exceeds that of heme, porphyrin intermediates accumulate, and "imbalanced" heme biosynthesis of increased porphyrin intermediates and decreased end product arises. Likely, to compensate for the reduced amount of heme synthesized, heme flux is increased (Fiorito et al, 2020). Cancer cells appear to have adopted a dependence on this seemingly "inefficient" metabolic pathway designed to produce aberrant levels of heme intermediates with unknown biological functions to date.

Early embryonic stem cells represent a special metabolic state that bears similarities with that of cancer cells (Smith & Sturmey, 2013). Therefore, we also studied the expression of the heme biosynthetic enzyme–encoding genes in human preimplantation embryos and embryonic stem cells. Specifically, we explored a high-resolution dataset generated upon single-cell RNA sequencing in human early embryonic cells and embryonic stem cells (Yan et al, 2013; Li et al, 2017). In the human zygote and early preimplantation embryos, the level of *ALAS1* expression is 100-fold higher than that of *FECH* (Fig 1C) (adjusted *P* < 0.001). This large difference in gene

expression for two key enzymes in heme biosynthesis was observed in all 17 early human embryos examined (Table S6). The large difference in expression levels is restricted to very early embryos because it rapidly narrows, via an increase in *FECH* expression and a decrease in *ALAS1* expression after initial rounds of embryonic cell division. *ALAS2* expression is not detected in any of the early human embryos, which is congruent with the non-essentiality of *ALAS2* in all examined cancers. As a control, the genes *AURKA* and *SOX4* show expected expression patterns during early embryogenesis (Penzo-Méndez et al, 2007; Sasai et al, 2008). The similarity of these gene expression data to those observed in cancer cells supports the possibility that a form of porphyrin overdrive operates in preimplantation human embryos.

## Porphyrin overdrive is absent in normal cells

To evaluate whether porphyrin overdrive is specific to cancer cells (and possibly preimplantation embryos), we investigated and compared heme biosynthesis in diverse types of non-cancerous cells, that is, normal differentiated cells, replicating fibroblasts, human primary hepatocytes, and human primary hematopoietic stem cells. We looked for evidence of porphyrin overdrive, defined by the non-homeostatic, imbalanced heme biosynthetic pathways and heightened accumulation of heme pathway intermediates. As existing literature predominantly identifies PPIX as the major accumulated form in both in vitro and in vivo settings (Szeimies et al, 1995; Moesta et al, 2001; Bellini et al, 2008; Kaneko & Kaneko, 2016), in this study, we refer to PPIX as the representative porphyrin species. Future studies will conduct a more detailed biochemical analysis to identify specific heme intermediate forms in different cancer cell types.

Using published datasets, we found that porphyrin overdrive is absent in normal differentiated cells and somatic stem cells (Fig 2A). First and consistent with this finding, normal human PBMCs do not accumulate heme intermediates even upon ALA induction (Oka et al, 2018). Second, colon cancer cells incubated with ALA to induce protoporphyrin IX (PPIX) production to accumulate PPIX, whereas no significant PPIX accumulation is observed in replicating human colon–derived fibroblasts (stromal cells) previously incubated with ALA (Krieg et al, 2002). Furthermore, the increased HMBS activity in colon cancer cells relative to stromal cells (Krieg et al, 2002) is consistent with a buildup of heme intermediates including PPIX. Third, using our previously described primary human hepatocyte system (Maher et al, 2020), we demonstrate that normal, metabolically active, primary hepatocytes do not accumulate porphyrins as indicated by the absence of the PPIX fluorescence (Fig 2B). In contrast, >99% of liver cancer cells produced and accrued PPIX upon induction with ALA, demonstrating a canonical feature of porphyrin overdrive.

To uncover the role of heme metabolism in normal human stem cells, we analyzed the erythropoiesis data generated by RNAi-based knockdown of gene expression in human hematopoietic progenitor cells (Egan et al, 2015). By mapping normal stem cell gene essentiality following the methods described in Egan et al (2015), we show that erythropoiesis is heme-dependent in both early progenitor (undifferentiated) cells and differentiated cells from normal human bone marrow–derived hematopoietic stem cells. In

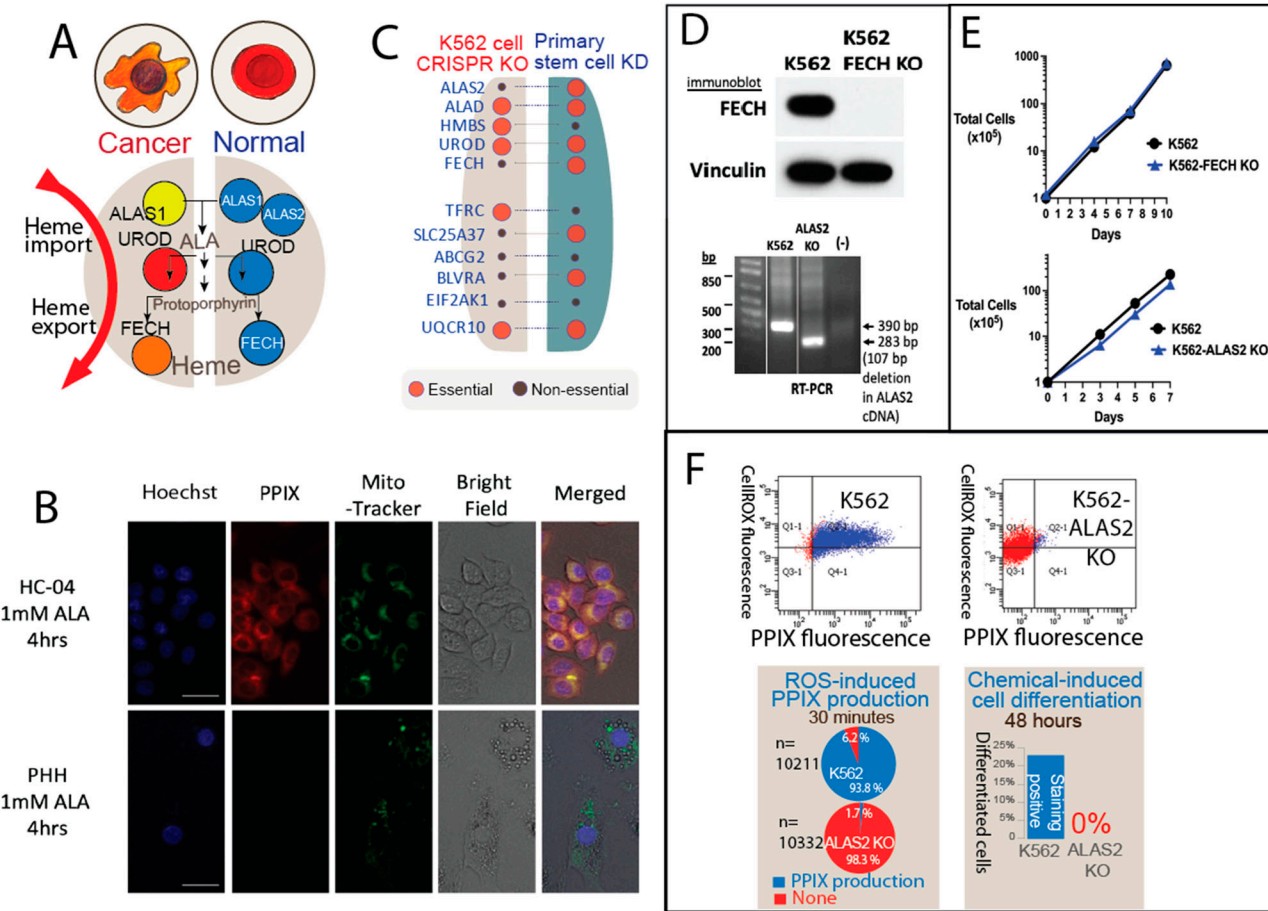

**Figure 2. Porphyrin overdrive is unique to cancer cells and absent in normal cells.**
**(A)** Schematic illustrations of the key differences of heme metabolism in cancer versus normal cells. Red, orange, and yellow shadings represent cancer gene essentiality as in Fig 1B. Blue colors refer to the normal heme biosynthesis process. **(B)** PPIX autofluorescence (red) is detected in HC-04 liver cancer cells but not in primary human hepatocytes after ALA addition, indicating PPIX accumulation in cancer cells but not in normal cells. MitoTracker Green staining indicates viable cells. The scale bar size is 25 μM. Over 1,000 cells were assessed for each sample (representative cells shown). PPIX red was not observed in any normal cells. **(C)** Distinct heme metabolic gene essentiality in stem cell and cancer cells. (Gene essentiality is represented as lethality after loss/deletion of a specific gene.) Right: the human primary bone marrow stem cell gene knockdown essentiality data used in this study were sourced from Egan et al (2015). These RNAi knockdown data demonstrate that normal stem cells rely on the first- and last-step genes for survival, revealing a distinct pattern compared with cancer cells. Left: cancer CRISPR KO data are from pan-cancer essentiality analysis (DepMap). **(D)** Validation of the K562-FECH KO cell line generated by CRISPR/Cas9 gene editing. Immunoblots of wild-type K562 and K562-FECH KO whole-cell lysates show a complete loss of FECH protein in K562-FECH KO cells, with vinculin used as a loading control (top). A 107-bp out-of-frame CRISPR-induced deletion was identified in the *ALAS2* gene (not shown), and because of the inability to identify a specific ALAS2 antibody for immunoblotting, the presence of this deletion was confirmed in RT–PCR analysis of K562-ALAS2 KO cells (bottom). **(E)** Total viable K562 and K562-FECH KO cells (top) and K562 and K562-ALAS2 KO cells (bottom) were determined over time using trypan blue exclusion, and no growth differences were detected. **(F)** ALAS2 KO K562 cells are arrested in a non-differentiable state. K562 cells readily differentiated upon ROS induction (tert-butyl hydroperoxide) and are committed into erythroid lineage within 48 h, as measured by activation of heme production and benzidine stain as a marker of differentiation. In contrast, ALAS2 KO cells do not respond to the ROS inducer (the first step of differentiation) nor commit to erythroid differentiation (parent versus ALAS2 KO, Mann–Whitney test, *P* < 0.001).

contrast to cancer cells, where *ALAS2* is dispensable, normal human hematopoietic stem cells depend on *ALAS2* for survival (Fig 2C) (Fig S7). The differential gene essentiality profiles between cancer and stem cells are consistent with the molecular basis for specific ex vivo purging of cancer hematopoietic progenitor cells, via ALA induction/photodynamic therapy (PDT) as a way to eradicate malignant cells while sparing hematopoietic stem cells (de Lima & Shpall, 2004). Taken together, the distinct genetic essentialities and the markedly different porphyrin accumulation modes between cancer and non-cancerous cells indicate that porphyrin overdrive is absent in somatic stem cells.

## CRISPR/Cas9-mediated KO of ALAS2 and FECH validates abnormal heme metabolism underlying cell proliferation and oncogenesis

We modified chronic myeloid leukemia K562 cells with CRISPR/Cas9-mediated KO of the genes encoding FECH and ALAS2 (Fig 2D). Under a heme homeostasis model, a cell line without *FECH* should not survive, because *FECH* is a single-copy gene in humans and with no functional substitute (Ponka et al, 2014). Indeed, knockdown of *FECH* in human stem cells led to cell death (Egan et al, 2015). However, under porphyrin overdrive conditions, we predicted that cancer cells rely on heme trafficking and are addicted to the

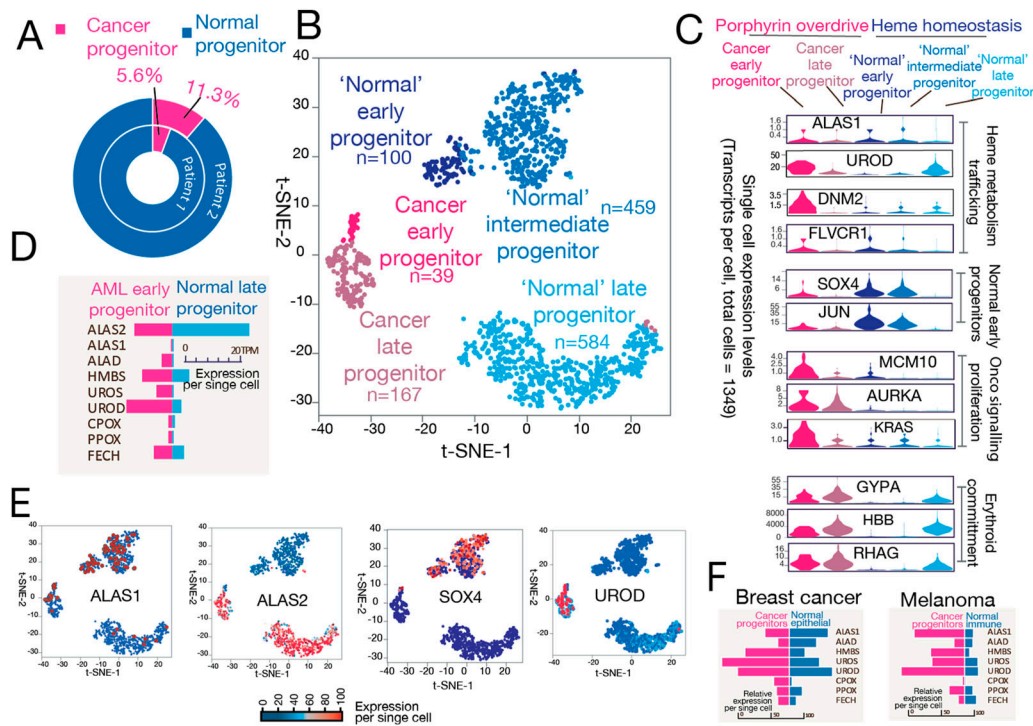

**Figure 3. Single-cell RNA sequencing of human AML bone marrow samples supports porphyrin overdrive as a hallmark of cancer cells.**
Single-cell transcriptomes were obtained from bone marrow biopsies of AML donors with over 60% blasts. **(A)** Single-cell population composition shows only a minor fraction of the cells are cancer progenitors, which are defined by transcriptome embedding and clustering analysis. **(B)** Identification of distinct single-cell populations in the AML patient bone marrow samples by t-SNE (t-distributed stochastic neighbor embedding) analysis. A total of 1,349 high-quality transcriptomes from patient biopsies were used. **(C)** Representative genes for proteins associated with heme metabolism, normal early erythroid development, cell proliferation, and erythroid commitment process are plotted. **(D)** UROD, the gene encoding the fifth enzyme of the heme biosynthetic pathway, is highly expressed in cancer early progenitors. HMS and UROS, which encode other intermediate step-catalyzing enzymes, are also overexpressed in cancer progenitors. **(E)** Marker genes delineate cell populations from the patient samples. Note the complementary expression patterns of ALAS1 versus ALAS2. **(F)** Solid tumor single-cell RNAseq shows the overexpression of the genes for the enzymes responsible for the intermediate heme biosynthetic pathway steps in breast cancer and melanoma (Tirosh et al, 2016; Wu et al, 2020).

porphyrin intermediates, and thus do not depend on the final step of heme biosynthesis for survival. As hypothesized, K562 cells with the *FECH* edited out (K562-FECH KO) (Fig 2D) had normal growth (Fig 2E) under standard culture conditions. The unimpeded growth of the K562-FECH KO cells presumably indicates that at least some cancer cell lines can metabolically function as heme auxotrophs.

As predicted, K562 cells with the *ALAS2* deleted (K562-ALAS2 KO) (Fig 2D) lost their differentiation capacity although their growth was not hindered (Fig 2E and F). Given the proposal that oxidation stress, with the involvement of reactive oxygen species (ROS), is the first step in erythroid differentiation of K562 cells (Chenais et al, 2000), we used the organic peroxide tert-butyl hydroperoxide (t-BHP) to trigger ROS-mediated erythroid differentiation. As early as 30 min after t-BHP–mediated ROS induction, the parental K562 cells started synthesizing heme, an early step in cellular differentiation (Chenais et al, 2000), whereas the K562-ALAS2 KO cells failed to initiate this metabolic step, indicated by the lack of heme production (Fig 2F) (Chenais et al, 2000). Furthermore, at 48 h, the parental K562 cells readily differentiated into erythroid cells, but the K562-ALAS2 KO cells continued to proliferate and failed to differentiate (Fig S8), as expected because ALAS2 is required for erythroid differentiation (Fig S7). Conceivably, in the absence of endogenous production of heme destined to hemoglobin, the

erythroid lineage commitment is blocked, forcing the K562-ALAS2 KO cells to an obligatory porphyrin overdrive path and into a proliferative/non-differentiable, or cancerous, state (Fig S8).

## Single-cell transcriptomics reveals porphyrin overdrive in acute myeloid leukemia (AML) patient biopsies

To study porphyrin overdrive in primary cancer cells, we performed single-cell RNAseq on biopsies from AML patients (Tables S7 and S8). For each patient, we obtained around 2,000 single-cell transcriptomes, and we analyzed cells of various differentiation stages and cell types. Interestingly, in two patients, only a minority of cells (5–10% of total cells) were identified as cancer progenitor cells with hyperproliferation features (Fig 3A), and they exhibited hybrid features of mature erythroid cells (e.g., expressing the genes HBB and GYPA for hemoglobin β and glycophorin A, respectively) and stem cells (e.g., expressing SOX4). The cancer progenitor cells (Fig 3B) had both mature cell features, for example, HBB expression, and early progenitor properties, for example, proliferative phenotype. Specifically, they were characterized by elevated iron and heme metabolic gene expression, cell proliferation markers, KRAS, and anabolism process genes, yet they had reduced normal stem cell progenitor marker gene expression, for example, SOX4 and JUN (Fig 3C) (Merryweather-Clarke et al, 2016). In addition, cancer/AML

progenitor cells exhibited imbalanced heme biosynthesis, inferred and exemplified by elevated gene expression for enzymes that catalyze the intermediate steps of heme biosynthesis, for example, HMBS and UROD (Fig 3D and E). The cancer progenitor cells also had a higher expression level of genes related to heme trafficking such as *FLVCR1* and *DNM2* (Fig 3C). Our results show that these cancer progenitor cells exhibit features of porphyrin overdrive, consistent with the abnormal heme anabolism and endogenous PPIX accumulation upon ALA administration in leukemia cells compared with normal PBMCs (Oka et al, 2018). Notably, in two solid tumors (Fig 3F) with independently generated single-cell transcriptomes (Tirosh et al, 2016; Wu et al, 2020), the genes encoding the enzymes for the intermediate steps of heme biosynthesis were similarly overexpressed in the cancer progenitor cells (Fig 3F).

From our single-cell transcriptome results, we searched for clues of specialized oncogenic microenvironments computationally that could support enhanced substrate trafficking and molecular interactions. Interestingly, only the cancer progenitor cells highly express cell-to-cell interaction-related genes, for example, *ICAM4*, *MAEA*, *ITGA4* (Fig S9A), which are critical in establishing the stem cell niche for erythropoiesis (Manwani & Bieker, 2008). We infer that the specific expression of these niche interaction genes suggests that the cancer progenitor cells exist in a specialized niche with intimate contact with neighboring cells and extracellular matrix, and establish evidence for future characterization of this unknown microenvironment.

Expression was remarkably enhanced for genes encoding proteins involved in metabolic substrate trafficking processes in the cancer progenitor cells, including genes for key protein players in lipid import purine and pyrimidine salvage pathways (Fig S9B). These specific metabolic flux–related genes, such as *DNM2* (endocytosis), *APOC1* (lipid transport), *MRI1* (L-methionine salvage), *MTAP* (NAD salvage), and *SLC29A1* (nucleoside import), indicate that the cancer progenitor cells represent hubs of metabolic trafficking activities. In particular, the lipid import gene *APOC1* is expressed in cancer progenitor cells, but not in the other cell populations, including normal progenitor cells, from the same patient samples (Fig S10A and B). The metastatin S100 family members linked to tumor niche construction (Lukanidin & Sleeman, 2012) are also enriched in the progenitor cells. These potential dynamic metabolic trafficking patterns inferred from the abundant expression of metabolite import genes in cancer early progenitors (Fig S9C) are reminiscent of "metabolic parasitism" (Icard & Lincet, 2013), a bioenergetically favorable process that uses energy-rich and preexisting macromolecules to fuel robust cell proliferation.

### Targeting cancer porphyrin overdrive with a bait-and-kill strategy

Two aspects of cancer porphyrin overdrive offer opportunities for therapeutic intervention. First, an imbalanced heme biosynthetic pathway (i.e., with aberrantly increased and suppressed enzyme activities) is absent in normal cells. Second, porphyrin overdrive is inducible 100-fold to 1,000-fold in cancer cells, but not in noncancerous cells. In fact, PDT, which uses ALA to induce biosynthesis and accumulation of photosensitizing porphyrins, and exemplifies the inducible principle of porphyrin overdrive, has been used for cancer imaging and treatment for decades (Krammer & Plaetzer,

2008). Thus, porphyrin overdrive may be a unique metabolic reprogramming feature of cancer cells that can be therapeutically exploited. Selective cancer cell death could possibly be achieved by activating PPIX-induced cytotoxicity with a two-step "bait-and-kill" strategy.

Knowing that cancer cells readily take up the heme precursor ALA and bypass the first and rate-limiting step of porphyrin biosynthesis catalyzed by ALAS, we set to assess our engineered two-step process for cancer metabolic death. This two-step process involves, first, to "bait" cancer cells with ALA to induce PPIX accumulation, and, second, to "kill" the cancer cells with a compound that exploits the metabolic stress associated with porphyrin accumulation. As expected, ALA addition to the culture medium did enhance PPIX buildup in K562 (Fig 4A) and HEL (Fig 4B) cells, as demonstrated by the few 100-fold to 1,000-fold PPIX concentration increase in relation to respective cells cultured in the absence of ALA (Fig 4A and B). Furthermore, HEL cell viability was not affected by supplementing the culture medium with ALA (1 mM final concentration), or ALA and DMSO (1 mM and 0.1% final concentrations, respectively) (Fig 4C). These results validated that the experimental criteria had been met to proceed with the second step of the "bait-and-kill" strategy.

To exploit the metabolic stress associated with porphyrin accumulation (e.g., oxidative stress, lipid peroxidation), we treated the human erythroleukemia HEL cell line with a range of concentrations of RSL3, an inhibitor of the cells' endogenous antioxidant defense system that protects cells against membrane lipid peroxidation (Cheff et al, 2023). As such, RSL3 induces ferroptosis, an iron-dependent form of programmed cell death highly relevant to iron and heme metabolism. Pretreatment of HEL cells with ALA increased their sensitivity toward RSL3 at least 1,000-fold (Fig 4D). Even at a RSL3 concentration of 100 mM, the viability of HEL cells remained similar to that of cells cultured in the absence of the RSL3 inhibitor. However, the viability of HEL cells grown in the presence of ALA for 4 h diminished with RSL3, reaching non-viability at a concentration of 17 mM. The decline was yet more acute when the incubation time of the HEL cells with ALA was extended to 24 h, and the cells became non-viable with 16 nM RSL3. Similar strong synergistic cancer-killing results, using ALA as "bait," and RSL3 and/or artemisinin as "kill," were found in murine BaF2_Jak2mt leukemic cells and human liver cancer cell line HC04 (Fig 4E and F). In contrast, ALA pretreatment did not sensitize cells to ganetespib, a heat shock protein inhibitor with antineoplastic activity (Alexandrova et al, 2015), suggesting that ALA is not a general inducer or sensitizer of cell death. Specifically, at ganetespib concentrations greater than 33 mM, HEL cells persist viably regardless of being grown in the presence or absence of ALA (Fig 4G). Notably, a healthy human primary lung fibroblast does not show PPIX accumulation (not shown), nor have treatment toxicity under the same conditions to eradicate cancer cells (Fig 4H and I).

In conclusion, we report that the essentiality pattern of the genes for the enzymes of the heme biosynthetic pathway varies significantly between human healthy and cancer cells. With the genes for the third, fourth, and fifth enzymes being particularly critical for cancer cell proliferation, accumulation of porphyrin intermediates exceeds heme production. We propose that cancer cells, in an

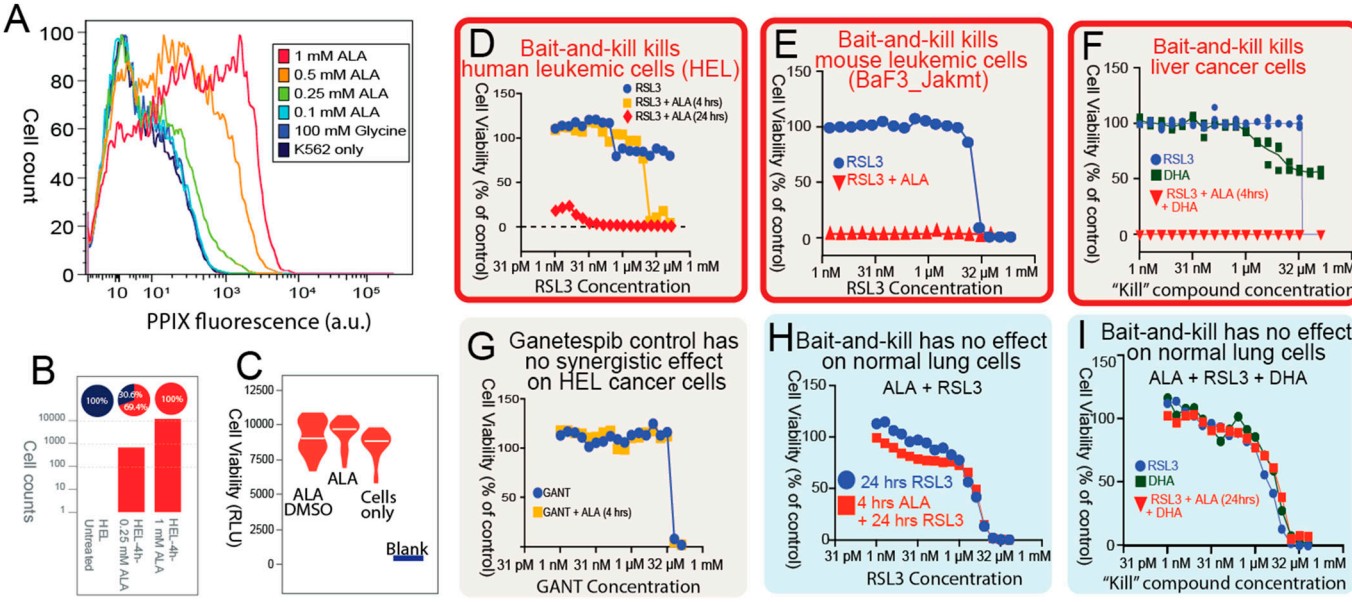

**Figure 4. Targeting porphyrin overdrive in cancer cells with a bait-and-kill strategy.**
**(A)** Concentration of PPIX accumulated in K562 cells depends on the ALA concentration supplemented with the culture medium. K562 cells were cultured in the absence or presence of 100 mM glycine or ALA (0.1, 0.25, 0.5, and 1.0 mM) at 37°C for 24 h, and PPIX fluorescence was measured by flow cytometry (a.u., arbitrary units) (0.1 mM versus 1 mM ALA, $P < 0.001$). **(B)** ALA-induced PPIX accumulation in HEL cells. PPIX accumulated in 100% of the HEL cells (n > 10,000 cells) as detected by PPIX fluorescence 4 h after treatment with 1.0 mM ALA (control versus 1 mM ALA, $P < 0.001$). The red sectors of the pie chart above the graph bars indicate the percentage of PPIX-accumulating HEL cells, whereas those in blue indicate the percentage of cells with no accumulated PPIX. The normalized PPIX values (%) indicated on the pie charts were obtained by dividing the PPIX fluorescence by the PPIX fluorescence intensity value for HEL cells incubated in 1.0 mM ALA-containing medium for 4 h, which was arbitrarily assigned 100%. **(C)** Cell viability is unaffected using ALA ("bait") as an inducer of PPIX accumulation. HEL cells were cultured in the absence or presence of ALA (1 mM), or ALA (1 mM) and DMSO (0.1%) at 37°C for 24 h, and their viability was calculated from the generated luminescence upon reaction with CellTiter-Glo. The assays were conducted in triplicate (RLU, relative luminescence units). **(D, E, F)** Dose-dependent response of cell viability for cells treated with increasing concentrations of drugs in three cancer cell lines. DHA, dihydroartemisinin. **(G)** Dose-dependent response of cell viability treated with ganetespib. **(H, I)** Normal human primary lung cells have no synergistic killing effect. Cell viability was determined after incubation with a wide concentration range of compounds either in the absence or presence of ALA (1.0 mM) for 4 or 24 h. (Note that the concentration of either RSL3 or ganetespib spans a range from pM to 150 mM.)

attempt to counteract unbalanced heme biosynthesis, scavenge heme from their environment and deploy vigorous heme trafficking strategies. We designate the combined non-homeostatic heme synthesis and enhanced heme trafficking as porphyrin overdrive. We suggest that porphyrin accumulation because of porphyrin overdrive and the resulting oxidative stress present a vulnerability of cancer cells that may be amenable to novel therapeutic strategies to specifically destroy cancer cells, as exemplified by our proof-of-principle results (Fig 4A–I).

# Discussion

In our study, we revealed the "*double life*" of porphyrins, demonstrating their essential role of viability while also sensitizing cells to oxidative stress. The intriguing nature of this apparent paradox highlights the complexity of porphyrin overdrive in cancer, and we believe unraveling this biology is crucial for a comprehensive understanding. Our proposal suggests that porphyrins may play an as-yet-uncharacterized role in oncogenesis without exerting detrimental effects on tumors. This hypothesis is substantiated by the observation that high-grade or metastatic tumors exhibit endogenous PPIX production, even without ALA treatment, implying an intrinsic oncogenic role (Moesta et al, 2001; Bellini et al, 2008). This hypothesis suggests that the endogenous role of porphyrins in

oncogenesis is distinct from their role in being exploited (Algorri et al, 2021) for tumor killing in PDT.

The significantly low gene essentiality of ALAS1 in cancer cells, given its crucial role as an indispensable precursor in heme biosynthesis and its established essentiality for early embryogenesis in the Alas1 KO study (Okano et al, 2010; Lian et al, 2018), raises intriguing questions. This observed low essentiality of ALAS1 in cancers suggests a potential trafficking route for heme precursors, implying an active tumor microenvironment. We are actively exploring extracellular ALA transport mechanisms, specifically PEPT1/2 (McNicholas et al, 2019; Harada et al, 2022). The ensuing discussion on these transport mechanisms has the potential to provide a plausible explanation for the observed low essentiality of ALAS1 in cancer cells, paving the way for our future studies.

The mechanism underlying the enhanced antitumor activity of RSL3 in the presence of overproduced porphyrins, especially without light irradiation, remains unknown. This drug-based approach is distinct from existing sensitizing therapies with ALA, such as thermal or sonodynamic cancer treatment (Gong & Dai, 2021; Bunevicius et al, 2022). To shed further light on this, we propose that RSL3 functions as a modulator, weakening the cell's endogenous antioxidant system (Cheff et al, 2023) that is needed for handling high levels of porphyrins. Our hypothesis suggests that the accumulation of PPIX enhances redox vulnerability, leading to cell killing without the need for light. This phenomenon shares similarities with the dark toxicity of aberrant porphyrin accumulations in

internal organs, as studied in the field of porphyria (Morehouse & Mason, 1990; Maitra et al, 2019). Targeting these vulnerabilities in cancer metabolism could provide new avenues for therapies.

# Materials and Methods

### Cell culture

The human chronic myelogenous leukemia cell line K562 (Cat No. CCL-243; ATCC) and human erythroleukemia cell line HEL 92.1.7 (Cat No. TIB-180; ATCC) were obtained from the American Type Culture Collection. Both the cells were grown in RPMI 1640 medium (Gibco) supplemented with 10% FBS (Sigma-Aldrich), gentamicin (1,000x, Cat No. 15-710-072; Thermo Fisher Scientific), Pen-Strep-Neo solution (100x, Cat No. 15640-055; Thermo Fisher Scientific), and 2 mM L-glutamine at 37°C in a humidified 5% $CO_2$ atmosphere. 5-Aminolevulinic acid hydrochloride (ALA), purchased from Alfa Aesar (Cat No. A16942ME), was dissolved in distilled water to yield a stock concentration of 1.0 M, and stored at –20°C. Glycine, purchased from Thermo Fisher Scientific (Cat No. BP381-500), was dissolved in phenol red–free culture medium purchased from Gibco (Cat No. 11835055) to give a stock concentration of 1 M.

### Cell growth and quantification

K562 or HEL cells were plated at $1-2 \times 10^5$/ml, and cells were counted by trypan blue (Cat No. 25-900-CI; Corning) exclusion over time, with passage back to the starting cell density as needed. Total cell numbers were determined based on the passage dilution at each time point, on a disposable hemocytometer (Cat No. DHC-N01; INCYTO).

### Generation of KO cell lines by CRISPR

K562 cells containing deletions in the ALAS2 or FECH genes were created using a multi-guide strategy via nucleofection (Lonza) of Cas9/RNP complexes (Gene KO Kit v2; Synthego) following the manufacturer's instructions. The guide RNAs used were as follows: ALAS2: UGAAGGCUUUCAAGACAGGU, CAAUCUUGCUCUUUCCCAUCC, and AGAUUCUCCAUCUUGGGCGA; FECH: UUAGACUCAUACCUCUUCUG, CUGGGCUGUUUCUGUGGUGA, and CUGACAGACCCUCCAGCUGC. CRISPR/Cas9 deletions were confirmed after amplification of the genomic region of interest, Sanger sequencing of the amplified region, and Inference of CRISPR Edits analyses (Synthego). Cells were lysed in RIPA buffer with protease and phosphatase inhibitors and immunoblotted with antibodies that recognize FECH (SC-377377) and vinculin (SC-73614) (Santa Cruz Biotechnology). PCR amplification of the first-strand cDNA from K562 and K562-ALAS2 KO was used to confirm the presence of the homozygous 107-bp out-of-frame deletion that was detected in the Inference of CRISPR Edits analyses of the edited genomic region in K562-ALAS2 KO cells.

### In vitro erythropoiesis protocols

In vitro erythropoiesis was monitored by following cellular differentiation of the cell lines (K562 and K562-ALAS2 KO) with benzidine staining. The benzidine solution was prepared by mixing 5 ml of 30% hydrogen peroxide ($H_2O_2$) with 1 ml of 0.2% benzidine dihydrochloride in 0.5 M acetic acid. Around one million cells were washed thrice with Dulbecco's phosphate-buffered saline (DPBS, 1X, Cat No. 21-031-CV; Corning) before being resuspended in 250 $\mu$l of 1x DPBS mixed with 250 $\mu$l benzidine solution and incubated at room temperature for 10 min. The cells stained brown-blue were visually recognized and counted as positives, on a disposable hemocytometer (Cat No. DHC-N01; INCYTO). The experiments were conducted in triplicates, and multiple microscopic fields were counted.

### AML patient single-cell RNAseq and heme biosynthetic pathway–related gene expression analysis

Cells were obtained from human donors with over 60% blast expansions in marrow biopsies. Blast cells were washed and isolated according to the standard tissue-banking protocols. Cells were carefully washed in DPBS (1X, Cat No. 21-031-CV; Corning) and resuspended at $10^6$ cells/ml to avoid cell aggregates. Cells were processed using the 10x Genomics Chromium controller and the Chromium Single Cell 3′ Library and Gel Bead kit (10x Genomics, Cat No. PN-1000075) following the standard manufacturer's protocols. First, gel beads-in-emulsion (GEMs) were generated by combining barcoded single-cell 3′ v3 gel beads, a master mix containing cells, and partitioning oil onto Chromium Chip B. To achieve single-cell resolution, cells were delivered at a limiting dilution, such that most (~90–99%) of the generated GEMs contain no cells, whereas the remainder contain predominantly a single cell. Between 2,000 and 21,000 live cells were loaded onto the Chromium controller to recover between 1,500 and 15,000 cells for library preparation and sequencing. Immediately after GEM generation, the gel beads were dissolved, primers were released, and any co-partitioned cells were lysed. An Illumina TruSeq Read 1 (read 1 sequencing primer), 16-nt 10x barcode, 12-nt unique molecular identifier, and 30-nt poly(dT) sequence were mixed with the cell lysate and a master mix containing reverse transcription (RT) reagents. Incubation of the GEMs produces barcoded, full-length cDNAs from polyadenylated mRNAs. Next, GEMs were broken, and cDNA was amplified and quantified using Agilent High Sensitivity DNA ScreenTape (Cat No. 5067-5592; Agilent Technologies). SPRIselect magnetic beads (Cat No. B23317; Beckman Coulter) were used to purify the first-strand cDNA from the post–GEM-RT reaction mixture, which included leftover biochemical reagents and primers. The barcoded, full-length cDNA was amplified via PCR to generate sufficient mass for library construction. Enzymatic fragmentation and size selection were used to optimize the cDNA amplicon size. TruSeq Read 1 (read 1 primer sequence) was added to the cDNAs during GEM incubation. P5, P7, a sample index, and TruSeq Read 2 (read 2 primer sequence) were added via end repair, A-tailing, adaptor ligation, and PCR. The final library quality was assessed using an Agilent Bioanalyzer high-sensitivity chip. Samples were then sequenced on the Illumina NextSeq 550 with a target of 150,000 reads/cell.

The Cell Ranger Single-Cell Software Suite (10x Genomics) was used for data processing, sample demultiplexing, and gene expression quantification. For data analysis, genes with more than one unique molecular identifier count were used. The top 1,000 most variably expressed genes were used for further clustering, and

t-distributed stochastic neighbor embedding analysis was performed to reduce the data dimension to a two-dimensional space, and k-means clustering was used to identify cell populations. The mean expression of genes in all cells in a given cluster was calculated, and the expression of each gene was compared with that of the same gene in all the other clusters. For cell classification, the mean expression profiles of all cells were first calculated, and then, each cell was assigned to a subpopulation by the highest Spearman correlation.

### Heme biosynthetic pathway–related gene expression analysis

The gene expression patterns related to heme biosynthesis from patient-derived tumor samples were analyzed using the publicly available resources to study tissue-specific gene expression, the GTEx project (Lonsdale et al, 2013; The GTEx Consortium et al, 2015), and TCGA program (Cancer Genome Atlas Research Network et al, 2013). The gene expression patterns of the heme biosynthetic pathway were investigated in a comprehensive analysis involving 10,000 tumors and matched normal tissues. This analysis covered a diverse range of major tumor types sourced from TCGA and GTEx projects. The examined tumor types included acute myeloid leukemia (LAML), adrenocortical carcinoma, bladder urothelial carcinoma, brain lower grade glioma, breast invasive carcinoma, cervical squamous cell carcinoma and endocervical adenocarcinoma, cholangiocarcinoma, chronic myelogenous leukemia, colon adenocarcinoma, controls, esophageal carcinoma, FFPE Pilot Phase II, glioblastoma multiforme, head and neck squamous cell carcinoma, kidney chromophobe, kidney renal clear cell carcinoma, kidney renal papillary cell carcinoma, liver hepatocellular carcinoma, lung adenocarcinoma, lung squamous cell carcinoma, lymphoid neoplasm diffuse large B-cell lymphoma, mesothelioma, miscellaneous, ovarian serous cystadenocarcinoma, pancreatic adenocarcinoma, pheochromocytoma and paraganglioma, prostate adenocarcinoma, rectum adenocarcinoma, sarcoma, skin cutaneous melanoma, stomach adenocarcinoma, testicular germ cell tumors, thymoma, thyroid carcinoma, uterine carcinosarcoma, uterine corpus endometrial carcinoma, and uveal melanoma. An index, reflecting the genes predominantly upregulated in cancers, was computed by calculating the paired differences between tumor and normal tissues within each major tumor type. Mann–Whitney's tests were performed, and $P$-values were adjusted for multiple testing using the Benjamini–Hochberg correction.

### CRISPR pan-cancer gene essentiality analysis

The data from the DepMap Portal were used to calculate the essentiality scores in a similar manner to published methods (Aguirre et al, 2016; Meyers et al, 2017; Kim & Hart, 2021). The whole-genome CRISPR/Cas9 datasets were used to identify significantly depleted mutant cells bearing a specific gene knockout in a pooled experiment. Gene essentiality was estimated from a given gene dependency inferred from CRISPR/Cas9 gRNA gene KO. The essential score was used to evaluate the cell growth fitness. The lower the essentiality score value, the larger the gene loss effect on cell viability. Thus, a score of 0, < 0, and > 0 indicates no fitness change, fitness loss, and fitness gain (i.e., possible growth advantage for the cell line) under the assay conditions, respectively. The method as

described in Aguirre et al (2016); Kim & Hart (2021) was employed to correct the copy-number bias in whole-genome CRISPR/Cas9 screens by computing the mean of sgRNAs versus the control plasmid library. Commonly essential genes were required for the fitness of most cell lines across cancer types (Dempster et al, 2019; Pacini et al, 2021). For in vivo gene essentiality analysis in pancreatic and lung cancer models, the published data in Zhu et al (2021) were used to specifically examine the genes encoding the heme biosynthetic pathway enzymes and heme transporters. We conducted Mann–Whitney's tests comparing the essentiality of the mid-step gene UROD with that of the first- and last-step genes. This analysis showed significant differences ($P < 0.001$) in essentiality patterns among different genes in the same biosynthetic pathways.

### In vitro primary human hepatocyte culture, quantification, and imaging

Sterilized 384-well plates (Cat No. 781091; Greiner) were unpackaged in a class II biosafety cabinet and placed in a secondary container (i.e., plates were placed in large assay pans) to serve as a lid and control evaporation. The day before hepatocyte seeding, wells were collagen-coated with 40 $\mu$l of 15 $\mu$g $\mu$l$^{-1}$ rat tail collagen I (Cat No. 354236; Corning) in sterile-filtered 0.02 M acetic acid (Cat No. AC456850050; Thermo Fisher Scientific), and kept at 37°C overnight. Immediately before seeding, the wells were washed thrice with sterile PBS and then filled with 20 $\mu$l in vitro GRO CP plate medium (Cat No. Z99029; BioIVT) supplemented with 1x Pen-Strep-Neo solution (100x, Cat No. 15640-055; Thermo Fisher Scientific) and 20 $\mu$M gentamicin (1,000x, Cat No. 15-710-072; Thermo Fisher Scientific). Vials of cryopreserved (male) primary human hepatocytes (Cat No. M00995-P; BioIVT) were thawed by immersion in a 37°C water bath for 2 min and sterilized with 70% ethanol in a sterile field, and the contents were added directly to 4 ml plate medium. Live and dead cells were quantified by trypan blue exclusion on a Neubauer-improved hemocytometer. The hepatocyte density was set at 1 × 10$^3$ live cells $\mu$l$^{-1}$, and 18 $\mu$l cell suspension was added to each well. Medium was exchanged with the GRO CP plating medium, described above, thrice weekly. The cells were incubated with 1.0 mM ALA at 37°C for 4 h. Both ALA-treated and non-treated cells were handled under very low light conditions. During the last 45 min of incubation, a staining solution diluted in phenol-free, serum-free RPMI (Cat No. 11835055; Gibco) containing Hoechst 33342 (Cat No. H3570; Life Technologies) at a final concentration of 10 $\mu$M was added to the cells. Live-cell imaging was performed on CellInsight CX7 High-Content Screening Platform (Thermo Fisher Scientific), and each plate well was counted for hepatocyte nucleus staining.

### In vitro hepatocyte cell line HC-04 culture, quantification, and imaging

Cryopreserved HC-04 hepatocyte cells were thawed, suspended into a previously prepared hepatocyte culture medium, and transferred to a T75 flask coated with collagen (Cat No. 354236; Corning) at 5 $\mu$g/cm$^2$. The previously prepared hepatocyte cell line culture medium consisted of a mixture of F12 base medium (Cat No. 11765-054; Invitrogen) and MEM base medium (Cat No. A10490-01; Invitrogen) on 1:1 (vol/vol) ratio, containing 10% FBS (Cat No.

SH30070; HyClone), 1.0 M Hepes (Cat No. 15630-080; Invitrogen), and 200 mM glutamine (Cat No. 25030-081; Invitrogen) (Sattabongkot et al, 2006). Cells were allowed to grow until reaching 70% confluence, and the medium was changed every other day. Then, the cells were trypsin-hydrolyzed with TrypLE Express Enzyme (1X) (Cat No. 12605028; Gibco) and washed with a hepatocyte culture medium. The HC-04 cells were seeded at a density of 6,000 cells/well and cultured in 384-well plates (Cat No. 781091; Greiner) in 20 $\mu$l of the above medium/well. Cells were incubated either in the absence or in the presence of 1.0 mM ALA at 37°C for 4 h. Both ALA-treated and non-treated cells were handled under very low light conditions. During the last 45 min of incubation, a staining solution diluted in phenol-free, serum-free RPMI (Cat No. 11835055; Gibco) containing Hoechst 33342 (Cat No. H3570; Life Technologies) at a final concentration of 10 $\mu$M was added to the cells. Live-cell imaging was performed on CellInsight CX7 High-Content Screening Platform (Thermo Fisher Scientific), and each plate well covering 15 fields at 20x was counted for hepatocyte nucleus staining.

### Cellular PPIX quantification

Protoporphyrin (PPIX) accumulation was determined using FACS as previously described (Fratz et al, 2014). K562 and K562-ALAS2 KO cells were independently seeded in 24-well plates for suspension culture (Cat No. 662102; Greiner) overnight at a density of $1.2 \times 10^5$ cells/well. The K562 and K562-ALAS2 KO cells were incubated in a medium (described in the Cell culture section) containing 1.0 mM ALA, at 37°C for 4 h. Preparation of either 5-ALA–treated cells or control (non–ALA-treated) cells was under very low light conditions. After incubation, cells were washed thrice with DPBS (1X, Ca$^{2+}$- and Mg$^{2+}$-free, Cat No. 21-031-CV; Corning) and resuspended in 250 $\mu$l of 1x DPBS. Briefly, cells were washed once with serum-free medium (Cat No. 11835055; Gibco) and incubated in six-well plates containing the medium (described in the Cell culture section) in either the absence or the presence of 100 mM glycine or ALA (0.1, 0.25, 0.5, and 1.0 mM) at 37°C. Intracellular PPIX concentration was measured 18 h later by FACS. The cell suspension was passed through a 40-$\mu$m Flowmi cell strainer to eliminate clumps and debris before transferring to BD Falcon tubes under very low light conditions to minimize phototoxicity caused by PPIX accumulation. FACS analyses were performed using BD LSR II Analyzer (Becton, Dickinson and Company) and FACSDiva version 6.1.3 software. To eliminate any background red fluorescence, the 633-nm red laser was blocked during the collection of the PPIX emission data. PPIX emission was determined in the 619-nm and 641-nm range (630/22 BP filter) upon excitation of the cells with the 405-nm laser. Forward-scatter versus side-scatter dot plots were used to gate the whole cells and thus remove the contribution of the cell debris from the population being examined. A minimum of 10,000 of the gated cells were then depicted in dot plots of side-scatter versus PPIX fluorescence; the gate was defined based on wild-type K562 cells without any perturbation as negative controls.

### Cellular ROS detection and cell viability assays

Both K562 and K562-ALAS2 KO cultures were seeded in 24-well plates for suspension culture (Cat No. 662102; Greiner) at a

density of $1.2 \times 10^5$ cells/well in the medium described in the Cell culture section, at 37°C overnight. After the incubation, cells were washed thrice with DPBS (1X, Ca$^{2+}$- and Mg$^{2+}$-free, Cat No. 21-031-CV; Corning), resuspended in 250 $\mu$l of 1x DPBS, and stained for 30 min at 37°C with 500 nM CellROX Green Reagent (Cat No. C10444; Life Technologies). During the last 15 min of staining, 1 $\mu$l of the 5 $\mu$M SYTOX Blue Dead Cell stain (Cat No. S34857; Life Technologies) was added to distinguish live from dead cells. Cells were immediately analyzed by FACS using BD LSR II Analyzer (Becton, Dickinson and Company) and FACSDiva Version 6.1.3 software. Emission of the oxidized fluorogenic CellROX Green was measured at 525 nm (530/30 BP filter) upon excitation of the cells using the 488-nm laser. K562 cells treated with 250 $\mu$M t-BHP (Cat No. 180340050; Thermo Fisher Scientific) for 15 min were used as a positive control. K562 cells were treated with 1 mM ALA for 4 h as a "PPIX fluorescence positive control." Both ALA-treated and non-treated cells are handled under very low light conditions.

### Drug ("bait-and-kill") assays

The compounds RSL3 (Cat No. HY-100218A; MedChemExpress) and ganetespib (Cat No. HY-15205; MedChemExpress) were dissolved in 100% DMSO (Cat No. 4-X; ATCC) to yield 10-mM stocks, which were stored at −80°C until further use. HEL cells were seeded in 38-well plates (Cat No. 781091; Greiner) at a density of 6,000 cells/well and in 20 $\mu$l of medium (described in the Cell culture section) per well. The cells were allowed to proliferate for the next 48 h. Both ALA-treated and non-treated cells were handled under very low light conditions throughout the assay. The cells were treated with 1 mM ALA in individual plates for 4 and 24 h. Both ALA-treated and non-treated plates were tested with the compounds RSL3 and ganetespib in triplicate wells using an 18-point concentration format with twofold dilutions (final concentrations of 132 $\mu$M to 1 nM) bringing the total volume to 25.6 $\mu$l per well. Cell proliferation was measured using the CellTiter-Glo 2.0 reagent (Cat No. G9243; Promega) to quantify cellular ATP according to the manufacturer's instructions by adding 25.6 $\mu$l of the reagent per well. Luminescence was measured with CLARIOstar Plus Microplate Reader (BMG Labtech). For each assay plate, a DMSO control (0.1%), a positive control, a negative control, and blanks were added, and a minimum of 12 wells per plate were analyzed. Data were reported as arbitrary luminometric units.

## Data Availability

The data used in the study are deposited at GSE263829, and the raw data files are available at https://www.ncbi.nlm.nih.gov/geo/query/acc.cgi?acc=GSE263829.

## Supplementary Information

# Acknowledgements

We thank Charles Szerekes for facilitating FACS analysis and the USF Genomics Program for sharing computing resources and facilities. Also, we thank Rudwan M Soukieh, Saeed Sinan, and Thomas D Nuhfer for discussions and for facilitating work related to this study. RHY Jiang, GC Ferreira, GW Reuther, and SM Sebti acknowledge support from the Florida Department of Health Grant (BHC 9BC14). RHY Jiang also acknowledges support from ACS-Moffitt (ACS-IRG IRG-14-189-19), University of South Florida (Women's Health Collaborative Grant and the College of Public Health), and the Gates Foundation (OPP1023601—for single-cell genomics protocol development).

## Author Contributions

SR Adapa: resources, data curation, formal analysis, validation, investigation, visualization, methodology, writing—original draft, and project administration.

GA Hunter: investigation, methodology, and writing—original draft.

NE Amin: investigation.

C Marinescu: investigation.

A Borsky: investigation.

EM Sagatys: methodology.

SM Sebti: investigation and writing—original draft.

GW Reuther: formal analysis, funding acquisition, validation, investigation, methodology, and writing—original draft.

GC Ferreira: data curation, supervision, funding acquisition, investigation, writing—original draft, and project administration.

RHY Jiang: conceptualization, data curation, formal analysis, supervision, funding acquisition, validation, investigation, visualization, methodology, and writing—original draft, review, and editing.

## Conflict of Interest Statement

The authors declare that they have no conflict of interest.

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
