## [Reviewer comments · Life Science Alliance]

Life Science Alliance

Porphyryn overdrive rewires cancer cell metabolism

Swamy Adapa, Gregory Hunter, Narmin Amin, Christopher Marinescu, Andrew Borsky, Elizabeth Sagatys, Said Sebti, Gary Reuther, Gloria Ferreira, and Rays Jiang

DOI: <https://doi.org/10.26508/lsa.202302547>

Corresponding author(s): Rays Jiang, USF Health Byrd Alzheimer's Institute

Review Timeline:

Submission Date:	2023-12-19
Editorial Decision:	2024-02-12
Revision Received:	2024-03-01
Editorial Decision:	2024-03-27
Revision Received:	2024-04-11
Accepted:	2024-04-12

Transaction Report:

February 12, 2024

Re: Life Science Alliance manuscript #LSA-2023-02547-T

Rays H.Y. Jiang
University of South Florida

Dear Dr. Jiang,

Thank you for submitting your manuscript entitled "Porphyrin overdrive rewires cancer cell metabolism" to Life Science Alliance. The manuscript was assessed by expert reviewers, whose comments are appended to this letter. We invite you to submit a revised manuscript addressing the Reviewer comments.

Thank you for this interesting contribution to Life Science Alliance. We are looking forward to receiving your revised manuscript.

Sincerely,

B. MANUSCRIPT ORGANIZATION AND FORMATTING:

Reviewer #1 (Comments to the Authors (Required)):

The manuscript "Porphyrin overdrive rewires cancer cell metabolism" by Rakesh et al., combines the use of large, published datasets, mainly CRISPR-based loss of function screens, with in vitro depletion and scRNAseq to interrogate to role of the porphyrin pathway in cancer metabolism.

Major Concerns

The manuscript utilizes data from several public sources. By itself, this is not concerning. However, the manuscript does not substantiate why given datasets are being used. For example, why start with metastatic cell lines and later primary tumors? The conclusions based on the in vivo Crispr screen appear to match what has already been published previously and the overall statistics of bringing together diverse expression databases are not well explained. Overall, it becomes difficult to judge what analysis has been performed by the authors and what is published data.

Throughout out the manuscript, it is often hard to judge whether statistics have been included to back claims.

The large number of references in the result section, often makes it unclear what experiments have been performed by the authors and further what the exact nature of the referenced studies are.

Minor Concerns

Line 89: How were the metastatic cell lines chosen? Are these the only metastatic lines in DepMap?

Line 91: "Significantly, these metastatic cancers developed 6 92 dependencies..." please do not use cancers and cell lines interchangeably. These are still cell lines.

Line 101: "This initial analysis suggests that cancer is dependent on imbalanced heme metabolism". Please specify.

Line 113: "In contrast to ALAS2, essentiality of ALAS1 is a feature of diverse cancer cells independent of cancer type (Fig. S2A)." Is this statement backed by statistics - the figure has no mention of any.

Line 117 and corresponding figure legend for S2B: "(B) A study of ~10,000 patient derived tumors from GTEx and TCGA shows that ALAS1 expression is elevated in tumors compared to normal cells, while like in normal cells, ALAS2 expression is absent in most cells)". Again, no statistics is provided to back the claim.

Line 125: "To better understand the molecular and genetic mechanisms underlying the cancer dependency on heme metabolism, we determined gene essentiality from genome-scale CRISPR/Cas9 loss-of-function screens in over 300 human cancer cell lines covering different cell lineages and estimated gene dependency". Please clarify why the initial screen focused on metastatic cancer lines, while now the focus is all cancer cell lines.

Line 174: "This analysis of the CRISPR/Cas 9 loss-of-function screens performed in mouse model systems in pancreatic and lung models revealed that aberrant heme metabolism has increased importance in both of these cancer types that were assessed" As far as I can see, this is a summary of the finding of the Zhu et al paper. If so, this cannot be a part of the result section.

Line 183: "the third enzyme of the heme biosynthetic 184 pathway, and the heme exporter FLVCR1 were among the most upregulated (Fig. S6A185 B) (Supplementary Table S5)." Table S5 provides data on select genes. Hence, it is not possible to review if this is amongst the most upregulated. Please provide a figure showing the results genome wide. This could be a volcano pot.

Line 250: "Using published datasets, we found that porphyrin overdrive, is absent in normal 251 differentiated cells and somatic stem cells (Fig. 2A)" Please expand figure legend and text to explain what was analyzed.

Line 261: Please include statistics on the FACS experiment.

Line 265: "To uncover the role of heme metabolism in normal human stem cells, we analyzed the erythropoiesis data generated by RNAi-based knockdown of gene expression". Figure 2C says the data is based on CRISPR?

Line 268: "we show that erythropoiesis is heme-dependent in both early progenitor (undifferentiated) cells and differentiated cells from normal human bone marrow-derived hematopoietic stem cells." I assume this is a re-analysis of the original dataset? If so, how was this done.

Line 297: "As predicted, K562 cells with the ALAS2 deleted (K562-ALAS2 KO) (Fig. 2D) lost their differentiation capacity although their growth was not hindered (Fig. 2E, F)." Based on the figure legend the differentiation capacity were assessed by benzidine stain. Please show the experimental data and include statistics.

Supplemental Figure S4: Please clarify the figure legend + figure.

Please include statistics in the methods section.

Reviewer #2 (Comments to the Authors (Required)):

In the manuscript, the authors indicated, based on available data, that cancer cells intrinsically exhibited overproduction of heme intermediates due to overexpression of HMBS, UROS, and UROD and increased heme flux; this condition is designated "porphyrin overdrive" even without ALA treatment, and increased heme flux. The authors addressed from the data from CRISPER pan-cancer gene essentiality analysis that porphyrin overdrive is essential for cancer cell viability. The authors found that early embryonic cells also showed "porphyrin overdrive" like in cancer cells and demonstrated by their own data that AML cells also exhibited "porphyrin overdrive". A bait-and-kill strategy called by the authors using RSL3, an inhibitor of GPX4, which can induce lipid peroxidation, sensitized the anti-tumor activity of RSL3 against HEL cells.

Major comments

It's important to distinguish between the accumulation of porphyrins and porphyrinogens in cancer cells exhibiting "porphyrin overdrive." Data supporting this distinction would strengthen the argument and clarify the biochemical landscape underpinning your findings.

The manuscript discusses how accumulated porphyrins can sensitize cancer cells to oxidative stress, yet also posits porphyrin overdrive as essential for viability. Elaborating on how these seemingly contradictory roles of porphyrins coexist in cancer cells would provide deeper insight into their complex biology.

It is very strange that the gene essentiality of ALAS1 in cancer cells is remarkably low because ALA is an indispensable precursor of heme biosynthesis, and a previous Alas1 knockout study suggested that Alas1 is essential for early embryogenesis (Okano et al. 2010). The low essentiality of ALAS1 raises questions about alternative heme production pathways in cancer. Discussion on extracellular ALA transport mechanisms, possibly via PEPT1/2, could offer a plausible explanation for this observation.

The mechanism by which overproduced porphyrins enhance the anti-tumor activity of RSL3, without light irradiation, remains unclear. Providing additional data or referencing relevant studies would be beneficial. Moreover, distinguishing your bait-and-kill strategy from existing sensitizing therapies with ALA such as thermal or supersonic cancer treatment, could highlight the novelty of your approach.

Minor comments

Scores of CRISPR based gene essentiality of ALAS1/2 in Sup Fig S2 are expressed differently from those in Sup Fig S1 and S3. Ensure consistency in the presentation of CRISPR-based gene essentiality scores across supplementary figures to aid in comparison.

Consider revising "the RSL3 inhibitor" in lines 402-403 to "the GPX4 inhibitor" or simply "RSL3" for accuracy.

Dear editor,

We appreciate the thorough review and constructive feedback provided by the reviewers on our manuscript titled "*Porphyrin Overdrive Rewires Cancer Cell Metabolism*." We are thankful that the editor and reviewers have taken their own time and effort to offer their suggestions and feedback. We have carefully considered each comment and made revisions to address the concerns raised. Below, we present a detailed response to each point raised by the reviewers. The reviewers remarks are in blue.

In response to Reviewer #1, we concur with the necessity to provide a more explicit clarification of our study subject and the statistical underpinning of our analyses. Given the heterogeneous nature of our work, presenting a substantial volume of data, we recognize the importance of offering clear explanations and robust statistical support for each data type. Our study draws from three main sources: **1)** Public data mining, encompassing CRISPR essentiality screens of cancer cell lines and extensive human tumor expression data; **2)** Previous data reanalysis, including our previous RNAi data on human primary hematopoietic stem cells and a published *in vivo* CRISPR metabolism gene study; and **3)** Newly generated data, involving primary human cells exhibiting the absence of porphyrin overdrive, CRISPR knockout validations in leukemic cell lines, studies on erythroid progenitor cell differentiation, single-cell RNA sequencing of AML patient samples, and drug assays. In this revision, we have taken great care to elucidate each data source, method, and the statistical approaches applied.

In response to Reviewer #2, we acknowledge the surprising aspect of our results regarding the non-essential nature of ALAS1, the rate-limiting, first step, of the pathway, in many cancers, despite its known essentiality in normal cells and embryonic development. In fact, this result is contrary to our initial hypothesis outlined in our grant proposal before the work started, which anticipated ALAS1 as essential for cancers! Our *in vitro* and *in vivo* CRISPR analysis, along with our own validations, have compelled us to recognize the critical importance of mid-steps in cancer metabolism. We are actively engaged in investigating the underlying mechanisms. And incorporated discussions in the revision.

Reviewer #2's insightful observation prompts a consideration of why cancers exhibit an overproduction of porphyrins. We propose that porphyrins might play an as-yet-uncharacterized role in oncogenesis, without detrimental effects on tumors. Our hypothesis suggests that the endogenous role of porphyrin in oncogenesis, is distinct, from its role in being exploited for tumor killing in photodynamic therapy (PDT). This proposition is supported by numerous human tumor expression studies, many directly derived from surgical tissue, which demonstrate elevated endogenous porphyrin production in tumors and its correlation with tumor aggressiveness. With the goal of tumor eradication in mind, we aim to leverage this overproduction of porphyrins in developing therapeutic strategies, including our bait-and-kill approach. We agree we need to acknowledge the importance of distinguishing our simple redox chemo strategy (without light or ultrasound) from current therapies to show its novelty. In response, we

have thoroughly explored these critical points, offering a clearer and more detailed discussion of our findings.

In this revision, in response to the reviewer's feedback, we have added

Fig. S2C **2C.** We provides a more detailed contrast between the essentiality of ALAS1 and mid-step gene UROD in various cancer cell line types in vitro.

Fig. S6. **Genes upregulated across diverse tumor types in the TCGA project.**
We provide a volcano plot to support the pan-cancer upregulation of the mid step gene HMBS and heme/porphyrin exporter gene FLVCR1).

Fig. S7. **Assessment of gene essentiality in heme biosynthesis during ex vivo erythropoiesis in human primary hematopoietic stem cells.** We provide a volcano plot to support the conclusion that stem cell needs first and last step genes

Fig. S8. **Erythroid differentiation in in vitro erythropoiesis assays.** We display benzidine staining images illustrating the differentiation failure of the ALAS2 knockout mutant, contrasting with the successful differentiation observed in the parent strain.

Detailed Statistical Methods in Public Data Mining and Experimental Data. We have enhanced the statistical methods section, offering comprehensive details on both our public data mining procedures and the statistical approaches applied to our experimental data.

In-Depth Clarification and Discussions on Porphyrin Roles and Functional Context in Current Literature. Our manuscript now includes an in-depth clarification and thorough discussions on the roles of porphyrin and its functional context within the current literature. We aim to provide a better understanding of the implications and significance of our findings in the broader scientific landscape.

We have incorporated the necessary revisions into the manuscript (**highlighted in red** in the revised version), and provided a detailed point-by-point response below (**Reviewer's comments are in blue**):

Reviewer #1 (Comments to the Authors (Required)):

The manuscript "Porphyrin overdrive rewires cancer cell metabolism" by Rakesh et al., combines the use of large, published datasets, mainly CRISPR-based loss of function screens, with in vitro depletion and scRNAseq to interrogate to role of the porphyrin pathway in cancer metabolism.

We thank the review for summarizing our work.

Major Concerns

The manuscript utilizes data from several public sources. By itself, this is not concerning. However, the manuscript does not substantiate why given datasets are being used. For example, why start with metastatic cell lines and later primary tumors?

We appreciate the reviewer's comment. In hindsight, we recognize that our initial mention of metastatic cell lines may have caused confusion. In the revision, we have removed the description of metastatic and now simply state that we selected a set of diverse cell lines from various tissue origins. The rationale for this selection lies in the shared properties of these cell lines within the context of the imbalanced pathway under investigation.

We have provided additional explicit explanations regarding the selection of previous stem cell knockdown data and human tumor expression data to facilitate a clearer understanding of the porphyrin overdrive process.

The conclusions based on the in vivo Crispr screen appear to match what has already been published previously and the overall statistics of bringing together diverse expression databases are not well explained. Overall, it becomes difficult to judge what analysis has been performed by the authors and what is published data. Throughout out the manuscript, it is often hard to judge whether statistics have been included to back claims. The large number of references in the result section, often makes it unclear what experiments have been performed by the authors and further what the exact nature of the referenced studies are.

We appreciate the reviewer's diligence in scrutinizing our manuscript, and we agree that we need a more explicit clarification of our study subject and the statistical underpinnings of our analyses. Recognizing the heterogeneous nature of our work, which encompasses a substantial volume of data, we understand the importance of providing clear distinctions between analyses conducted by the authors and the utilization of published data.

Our study relies on three primary sources: 1) Public data mining, which includes CRISPR essentiality screens of cancer cell lines and extensive human tumor expression data; 2) Previous data reanalysis, incorporating our own RNAi data on human primary hematopoietic stem cells and a published in vivo CRISPR metabolism gene study; and 3) Newly generated data, involving primary human cells demonstrating

the absence of porphyrin overdrive, CRISPR knockout validations in leukemic cell lines, studies on erythroid progenitor cell differentiation, single-cell RNA sequencing of AML patient samples, and drug assays.

In this revision, we have taken meticulous care to elucidate each data source, method, and the statistical approaches applied. We introduced Supplemental Fig. S6. and Fig. S7., illustrating the genome-wide significance of chosen genes through volcano plots. Furthermore, we explicitly explained the statistical tests in several figures. These refinements are designed to enhance the transparency and understanding of our study's methodology

Minor Concerns

Line 89: How were the metastatic cell lines chosen? Are these the only metastatic lines in DepMap?

We value the input from the reviewer. Upon reflection, we recognize that our initial reference to metastatic cell lines may have been confusing. In the revised version, we have eliminated the mention of metastatic and now explicitly state that we chose a diverse set of cell lines from various tissue origins. This selection is grounded in the common characteristics exhibited by these cell lines within the context of the imbalanced pathway.

Line 91: "Significantly, these metastatic cancers developed dependencies..." please do not use cancers and cell lines interchangeably. These are still cell lines.

Yes, we acknowledge and appreciate the reviewer's critique that the terms 'cancers' and 'cell lines' should *not* be used interchangeably. We recognize the distinction, especially considering the different essentialities observed in vivo compared to in vitro in the manuscript. Moreover, our recent work (to be submitted) shows the significant role of the tumor microenvironment in porphyrin production. In the revised manuscript, we have carefully specified the use of 'cell lines' where the results are derived to accurately convey our findings.

Line 101: "This initial analysis suggests that cancer is dependent on imbalanced heme metabolism". Please specify.

We intended to convey that the CRISPR essentiality data indicated a greater dependency of these cancer cell lines on the mid steps of biosynthesis genes, as opposed to the first or last steps of the pathways. This observation suggests that the cancer pathways accumulate intermediates more than the end product, which is indicative of an 'inefficient' pathway due to imbalanced synthesis enzyme steps. This stands in contrast to normal heme biosynthesis, which is well-balanced, with no intermediate accumulation, and efficient conversion of substrates into products. We have revised the text accordingly.

Line 113: "In contrast to ALAS2, essentiality of ALAS1 is a feature of diverse cancer

cells independent of cancer type (Fig. S2A)." Is this statement backed by statistics - the figure has no mention of any.

We have performed a nonparametric Mann-Whitney test and demonstrated that the essentiality patterns of these two isoforms are distinct. In this revision, we have incorporated the details of this statistical analysis into the manuscript to provide a clear and explicit backing for the statement in question.

Line 117 and corresponding figure legend for S2B: "(B) A study of ~10,000 patient derived tumors from GTEx and TCGA shows that ALAS1 expression is elevated in tumors compared to normal cells, while like in normal cells, ALAS2 expression is absent in most cells)". Again, no statistics is provided to back the claim.

We appreciate the reviewer's attention to the necessary details. We have conducted Mann-Whitney tests, providing statistical support for the pairwise comparisons of gene expression patterns between tumors and normal cells for ALAS1 and ALAS2. In this revision, these results have been added to the manuscript to substantiate the claim made in the corresponding figure legend for S2B.

Line 125: "To better understand the molecular and genetic mechanisms underlying the cancer dependency on heme metabolism, we determined gene essentiality from genome-scale CRISPR/Cas9 loss-of-function screens in over 300 human cancer cell lines covering different cell lineages and estimated gene dependency". Please clarify why the initial screen focused on metastatic cancer lines, while now the focus is all cancer cell lines.

We appreciate the reviewer's observation. In the revision, we have removed the term 'metastatic' as it was not relevant and could lead to confusion. The initial analysis focused on a diverse subset of cell lines, and our rationale was to transition from studying a subset to a larger set. Throughout this transition, we consistently observed similar imbalances in heme biosynthesis pathways, supporting the broader focus on all cancer cell lines in the subsequent analyses.

Line 174: "This analysis of the CRISPR/Cas 9 loss-of-function screens performed in mouse model systems in pancreatic and lung models revealed that aberrant heme metabolism has increased importance in both of these cancer types that were assessed" As far as I can see, this is a summary of the finding of the Zhu et al paper. If so, this cannot be a part of the result section.

We acknowledge the reviewer's concern and appreciate the opportunity to clarify. Indeed, Zhu et al. have previously highlighted the importance of the heme pathway in pancreatic and lung cancer, attributing it to the scarcity of heme availability in the tumor microenvironment (TME) in vivo, necessitating increased heme production. In our study, we reexamined this data and specifically focused on demonstrating that an imbalanced pathway is operating in vivo. Notably, we observed that the first part of the pathway has low essentiality (an important detail not mentioned in Zhu et al.), while the mid-late part

of the pathway exhibits increased essentiality. Our intention was to illustrate that in vivo, tumors also depend on an imbalanced pathway to overproduce porphyrin. We recognize the need for clarity on this point and have refined the language in the revision to convey this distinction more explicitly.

Line 183: "the third enzyme of the heme biosynthetic pathway, and the heme exporter FLVCR1 were among the most upregulated (Fig. S6A185 B) (Supplementary Table S5)." Table S5 provides data on select genes. Hence, it is not possible to review if this is amongst the most upregulated. Please provide a figure showing the results genome wide. This could be a volcano plot.

We appreciate the reviewer's suggestion to employ a volcano plot for a comprehensive view of genome-wide significance. In response to this feedback, we have added Supplemental Figure S6, illustrating the genome-wide results. This figure confirms that the third step of the heme biosynthetic pathway (HMBS) and the heme exporter FLVCR1 are among the most upregulated genes. We find this result striking, implicating a common role for heme/porphyrin exporting in diverse tumors. This addition provides a visually accessible demonstration of the genome-wide significance of our findings, enhancing the clarity and robustness of our results.

Line 250: "Using published datasets, we found that porphyrin overdrive, is absent in normal differentiated cells and somatic stem cells (Fig. 2A)" Please expand figure legend and text to explain what was analyzed.

We appreciate the reviewer's request for clarification on the analysis presented in Figure 2A. In response to this feedback, we have expanded the figure legend and accompanying text to provide a detailed explanation. Specifically, we reanalyzed previous RNAi data on human hematopoietic stem cells, illustrating that normal stem cells require the first and last step genes for survival, unlike cancer cell lines. We have added a new volcano plot Supplemental Fig S7, to visually depict the high essentiality of these genes in normal human stem cells. The legend has been enhanced to offer a more comprehensive understanding of the analysis performed and the results presented.

Line 261: Please include statistics on the FACS experiment.

We appreciate the reviewer's request for statistical information on the FACS experiment presented. In response, we conducted both the chi-square test and Fisher's exact test to rigorously assess the significance of the FACS results. The details of these statistical analyses have been added to the manuscript.

Line 265: "To uncover the role of heme metabolism in normal human stem cells, we analyzed the erythropoiesis data generated by RNAi-based knockdown of gene expression". Figure 2C says the data is based on CRISPR?

We acknowledge the legend of Figure 2C lacks clarity, and we are grateful for bringing it to our attention. The data presented in Figure 2C is indeed derived from an RNAi knockdown experiment in human primary hemopoietic stem cells (Right), in contrast to the CRISPR KO in cancer cell lines (Left). We have promptly revised the figure legend to accurately reflect the experimental methodology.

Line 268: "we show that erythropoiesis is heme-dependent in both early progenitor (undifferentiated) cells and differentiated cells from normal human bone marrow-derived hematopoietic stem cells." I assume this is a re-analysis of the original dataset? If so, how was this done.

We appreciate the reviewer's inquiry. As mentioned in our response to the human hematopoietic stem cell RNAi KO, the original data provided the essentiality levels of genes in human stem cells. For the re-analysis, we re-conducted a genome-wide non-parametric test with BH p-value adjustment to derive statistically significant results. To provide a more comprehensive view, we have added a new Supplemental Figure S7 of a volcano plot to visually represent the results. Additionally, we have incorporated more details on the statistical methodology used in the revised manuscript to enhance transparency and understanding.

Line 297: "As predicted, K562 cells with the ALAS2 deleted (K562-ALAS2 KO) (Fig. 2D) lost their differentiation capacity although their growth was not hindered (Fig. 2E, F)." Based on the figure legend the differentiation capacity were assessed by benzidine stain. Please show the experimental data and include statistics.

We have included Supplemental Figure S8 to provide the staining results using benzidine, clearly demonstrating the failure of differentiation in ALAS2-KO cells. Additionally, we have incorporated relevant statistical information to enhance the robustness and interpretation of these experimental findings in the revised manuscript.

Supplemental Figure S4: Please clarify the figure legend + figure.

We appreciate the reviewer's request for clarification on Supplemental Figure S4. This figure is intended to illustrate the varying degrees of partial essentiality within the heme biosynthesis pathway across diverse cancer cell types. Despite differences in the extent of essentiality, the figure demonstrates that partial essentiality is a common feature. The heme biosynthetic pathway comprises eight enzymatic steps. The classification of porphyrin overdrive metabolism is determined by the number of essential enzyme-encoding genes within the heme biosynthetic pathway, as inferred from the gene essentiality results in cell lines. We have decided to remove this figure as it does not add significantly to our message.

Please include statistics in the methods section.

In response to this feedback, we have diligently added detailed statistical information regarding the testing of essentiality differences, and expression differences to the methods section. These additions aim to provide a comprehensive understanding of the statistical approaches applied in our study.

Reviewer #2 (Comments to the Authors (Required)):

In the manuscript, the authors indicated, based on available data, that cancer cells intrinsically exhibited overproduction of heme intermediates due to overexpression of HMBS, UROS, and UROD and increased heme flux; this condition is designated "porphyrin overdrive" even without ALA treatment, and increased heme flux. The authors addressed from the data from CRISPER pan-cancer gene essentiality analysis that porphyrin overdrive is essential for cancer cell viability. The authors found that early embryonic cells also showed "porphyrin overdrive" like in cancer cells and demonstrated by their own data that AML cells also exhibited "porphyrin overdrive". A bait-and-kill strategy called by the authors using RSL3, an inhibitor of GPX4, which can induce lipid peroxidation, sensitized the anti-tumor activity of RSL3 against HEL cells.

We appreciate the reviewer for providing a concise summary that accurately captures the key findings of our study.

Major comments

It's important to distinguish between the accumulation of porphyrins and porphyrinogens in cancer cells exhibiting "porphyrin overdrive." Data supporting this distinction would strengthen the argument and clarify the biochemical landscape underpinning your findings.

We agree with the reviewer and acknowledge the significance of distinguishing between the accumulation of porphyrins and porphyrinogens in cancer cells undergoing 'porphyrin overdrive.' Although existing literature predominantly points to PPIX as the major accumulated form in both in vitro and in vivo settings, we recognize the necessity for a more detailed biochemical analysis to identify specific heme intermediate forms in different cancer cell types, both in vitro and in vivo. Actively conducting a set of biochemical experiments, we aim to discern the various forms, and these insights will play a crucial role in shaping our future studies. We have included references for PPIX and refined our writing for clarity.

The manuscript discusses how accumulated porphyrins can sensitize cancer cells to oxidative stress, yet also posits porphyrin overdrive as essential for viability. Elaborating on how these seemingly contradictory roles of porphyrins coexist in cancer cells would provide deeper insight into their complex biology.

Reviewer #2's perceptive observation highlights the apparent contradiction in the roles of accumulated porphyrins in cancer cells, where they are both essential for viability and

can sensitize cells to oxidative stress. We acknowledge this apparent paradox and believe that unraveling this complex biology is crucial for a comprehensive understanding of porphyrin overdrive in cancer. Our proposal is that porphyrins might play an as-yet-uncharacterized role in oncogenesis without exerting detrimental effects on tumors. This hypothesis posits that the endogenous role of porphyrins in oncogenesis is **distinct** from their role in being exploited for tumor killing in photodynamic therapy (PDT). To support this proposition, we refer to numerous human tumor expression studies, many directly derived from surgical tissue, which demonstrate elevated endogenous porphyrin production in tumors and its correlation with tumor aggressiveness. In response, we have extensively delved into these crucial points, providing a clearer and more detailed discussion of our findings.

It is very strange that the gene essentiality of ALAS1 in cancer cells is remarkably low because ALA is an indispensable precursor of heme biosynthesis, and a previous *Alas1* knockout study suggested that *Alas1* is essential for early embryogenesis (Okano et al. 2010). The low essentiality of ALAS1 raises questions about alternative heme production pathways in cancer.

We appreciate the reviewer's remark. The remarkably low gene essentiality of ALAS1 in cancer cells, given its crucial role as an indispensable precursor in heme biosynthesis and its established essentiality for early embryogenesis in the *Alas1* knockout study, raises intriguing questions. To address this query, we are actively exploring extracellular ALA transport mechanisms. We recognize the importance of clearly conveying this aspect to enhance our understanding of heme metabolism in cancer and incorporated relevant insights into the revised manuscript.

Discussion on extracellular ALA transport mechanisms, possibly via PEPT1/2, could offer a plausible explanation for this observation.

We concur with the reviewer that this low essentiality of ALAS1 in cancers suggests a potential trafficking route for heme precursors, implying an active tumor microenvironment. In response to this inquiry, we are actively engaged in exploring extracellular ALA transport mechanisms, with a specific focus on PEPT1/2. The ensuing discussion on these transport mechanisms has the potential to furnish a plausible explanation for the observed low essentiality of ALAS1 in cancer cells, and pave ways for our future studies

The mechanism by which overproduced porphyrins enhance the anti-tumor activity of RSL3, without light irradiation, remains unclear. Providing additional data or referencing relevant studies would be beneficial. Moreover, distinguishing your bait-and-kill strategy from existing sensitizing therapies with ALA such as thermal or supersonic cancer treatment, could highlight the novelty of your approach.

We appreciate the reviewer's observation regarding the mechanism underlying the enhanced anti-tumor activity of RSL3 in the presence of overproduced porphyrins,

particularly without light irradiation. To provide further insight, we have included additional references illustrating how RSL3 functions as an inhibitor to weaken the cell's antioxidant system. Our hypothesis posits that the accumulation of PPIX enhances redox vulnerability, resulting in cell killing without the need for light. This phenomenon bears resemblance to the dark toxicity of porphyrins in internal organs, as studied in the porphyria field. We have incorporated these additional details into our revised manuscript to enhance clarity.

Minor comments

Scores of CRISPR based gene essentiality of ALAS1/2 in Sup Fig S2 are expressed differently from those in Sup Fig S1 and S3. Ensure consistency in the presentation of CRISPR-based gene essentiality scores across supplementary figures to aid in comparison.

We've revised the presentation of CRISPR-based gene essentiality scores across supplementary figures for consistency. Thanks for pointing this out.

Consider revising "the RSL3 inhibitor" in lines 402-403 to "the GPX4 inhibitor" or simply "RSL3" for accuracy.

Certainly, we agree with the clarity issue with the current description. In the revision, we have simplified it to just "RSL3."

March 27, 2024

RE: Life Science Alliance Manuscript #LSA-2023-02547-TR

Dr. Rays H.Y. Jiang
University of South Florida
3720 Spectrum Blvd
Tampa 33612

Dear Dr. Jiang,

Thank you for submitting your revised manuscript entitled "Porphyrin overdrive rewires cancer cell metabolism". We would be happy to publish your paper in Life Science Alliance pending final revisions necessary to meet our formatting guidelines.

- please make sure the author order in your manuscript and our system match and is correct
- please upload all figure files as individual ones, including the supplementary figure files; all figure legends should only appear in the main manuscript file
- please remove figures from the manuscript file
- please add ORCID ID for the corresponding author -- you should have received instructions on how to do so
- please add a Summary Blurb/Alternate Abstract and a Category to our system
- please add the Twitter handle of your host institute/organization as well as your own or/and one of the authors in our system
- please remove the "One-Sentence Summary:" section from the manuscript text
- please incorporate any points from the Conclusion section into the Discussion; we only allow a Discussion section
- please consult our manuscript preparation guidelines <https://www.life-science-alliance.org/manuscript-prep> and make sure your manuscript sections are in the correct order and labeled correctly -- separate results and discussion section
- please add your main, supplementary figure, and table legends to the main manuscript text after the references section
- please upload your tables in editable .doc or Excel files
- please add callouts for Figure S4A-B to your main manuscript text
- please indicate the scale bar size in Legend for Fig 2B
- please incorporate the Supplementary Methods into the main Materials and Methods section. We do not have a word limit for this section. Same should be done for the Supplemental References into the main Reference list.
- complete the Data Availability statement with accession numbers

A. FINAL FILES:

B. MANUSCRIPT ORGANIZATION AND FORMATTING:

Sincerely,

Reviewer #1 (Comments to the Authors (Required)):

I still think that "The rationale for this selection lies in the shared properties of these cell lines within the context of the imbalanced pathway under investigation" sounds a lot like cherry-picking datasets, but I will leave that up to the editor to decide.

At this point, I have no further concerns.

Reviewer #2 (Comments to the Authors (Required)):

The authors responded to all comments and I recommend that this paper be accepted.

April 12, 2024

RE: Life Science Alliance Manuscript #LSA-2023-02547-TRR

Dr. Rays H.Y. Jiang
USF Health Byrd Alzheimer's Institute
Molecular Medicine
3720 Spectrum Blvd
Tampa, FL 33612

Dear Dr. Jiang,

Thank you for submitting your Research Article entitled "Porphyrin overdrive rewires cancer cell metabolism". It is a pleasure to let you know that your manuscript is now accepted for publication in Life Science Alliance. Congratulations on this interesting work.

DISTRIBUTION OF MATERIALS:

Again, congratulations on a very nice paper. I hope you found the review process to be constructive and are pleased with how the manuscript was handled editorially. We look forward to future exciting submissions from your lab.

Sincerely,
